# LEARNING FREQUENCY DOMAIN CODES FOR SEMANTIC VISION

## ABSTRACT

Visually semantic concepts such as objects and categories provide a natural foundation for semantic reasoning, yet standard deep learning-based vision models routinely extract and aggregate features using homogeneous stacks of spatial layers. As a result, feature representations are learnt implicitly without clear organisation, rendering decision-making processes opaque and difficult to interpret. Psychovisual processing provides a way to mimic how the brain encodes and interprets visual information that produces higher abstractions from low-level processing. In this paper, we propose Semantic Visual Coding (SVC), a learnt frequency domain representation that introduces explicit psychovisual abstraction into convolutional neural networks (CNNs). Inspired by psychovisually motivated image codes from the 1990s, SVC learns band-limited filters that encode task-relevant semantics as distinct regions of the frequency domain. These converge towards sparse (data-driven) coronal patterns that suggest a natural representation scheme for semantic abstractions supporting model reasoning. We also introduce a framework that adapts CNNs to be psychovisually aware by combining traditional low-level spatial feature extraction with high-level abstraction in the frequency domain via SVC, which we call 'PsychoNet'. Salience analyses show that PsychoNet's spatial layers extract highly interpretable object parts and morphological features, unlike blob-like regions produced by standard CNNs. It further finds that SVC forms structured selections of these parts that are organised by spatial scale, suggesting frequency domain abstraction as a promising direction for interpretable models which reveal the semantic features they employ.

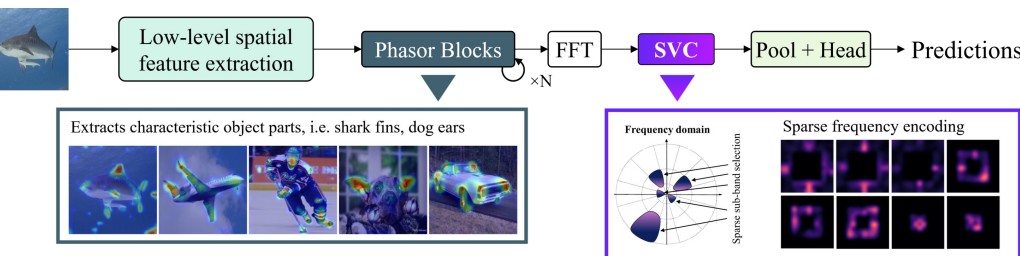

Figure 1: The brain encodes and interprets visual information using **psychovisual processing**, which separates feature extraction from higher cognition using intermediate abstractions. **PsychoNet** introduces similar pipelines to CNNs. Early spatial layers extract low-level features, similar to early cortical processing, and **Phasor Blocks** localise key characteristic object parts. Subsequently, these parts are encoded in the frequency domain by **Semantic Visual Coding (SVC)** into sparse frequency sub-bands. We believe that this is a naturally emergent representation for semantic information, similar to psychovisual abstractions. FFT denotes the Fast Fourier Transform.

## 1 INTRODUCTION

Ever since the ImageNet challenge popularised deep learning for computer vision (Krizhevsky et al., 2017; Deng et al., 2009), architectural advances have focused on the design of spatial domain feature extractors, from convolution layers (He et al., 2016; Xie et al., 2017; Huang et al., 2017; Liu

et al., 2022) to more recent token mixers based on attention mechanisms (Dosovitskiy et al., 2021; Tolstikhin et al., 2021; Gao et al., 2021; Rao et al., 2023). Although these models achieve impressive performance, the way they reason in deeper layers is often opaque and difficult to interpret. Psychovisual processing—the way human vision encodes and interprets visual information—separates feature extraction from higher cognition using intermediate abstractions, like objects, relations, and categories, providing a natural basis for reasoning (Quiroga et al., 2005; Kriegeskorte et al., 2008; Quiroga, 2012; Le et al., 2024). In contrast, current architectures refine and aggregate image features using homogeneous stacks of spatial layers, with the feature processing emerging implicitly through training and often lacking clear organisation.

In this work, we propose Semantic Visual Coding (SVC), a frequency-domain high-level processing module that bridges low-level feature extraction and decision making in CNNs, revealing interpretable, object-part-based semantic representations that underpin model reasoning within a psychovisual-inspired pipeline. Concretely, they are implemented with data-driven band-limited frequency filters, inspired by psychovisual coding schemes from the 1990s that targeted perceptually salient frequencies found by human vision studies (Saadane et al., 1994; Guedon et al., 1995; Saadane et al., 1998). Semantic Visual Coding (SVC) extends this idea to high-level abstraction by allowing networks to discover by learning the sparse frequency subsets most relevant to a given task. These promote richer intermediate abstractions grounded in task-oriented semantic features and yield potentially interpretable views of the model's reasoning, pointing to a naturally emergent framework for high-level abstraction with semantic reasoning. In future, we also aim to apply SVC to domains where the frequency domain is the natural measurement space, particularly magnetic resonance imaging (MRI) (Chandra et al., 2021) as transparency and trustworthy model behaviour are crucial due to the high-stakes nature of medical applications.

Additionally, we develop PsychoNet, a framework that incorporates SVC into conventional convolutional neural networks (CNNs), demonstrated on the widely used ResNet and state-of-the-art ConvNeXt architectures. PsychoNet establishes a coherent dual-domain pipeline (Figure 1): initial low-level image features are extracted in the spatial domain and augmented by learning complex-valued representations, then SVC constructs high-level abstractions in the frequency domain that support decision making with semantic reasoning. To the best of our knowledge, this work presents the first data-driven exploration of the frequency domain for high-level representation learning in vision, whereas prior studies focus mainly on lower level feature learning or parameterising spatial models (Chi et al., 2020; Rippel et al., 2015; Rao et al., 2023). To summarise, the key contributions of this work are:

- Inspired by psychovisual abstraction, we introduce SVC, a deep-learning based module that automatically learns frequency domain representations of high-level semantic image information for a given vision task. These emerge as sparse selections of coronal frequency sub-bands in the discrete Fourier Transform (DFT).

- Our PsychoNet (Figure 1) is a framework that integrates SVC into conventional CNN models. We demonstrate it enables interpretable psychovisual-like processing on ResNet and ConvNeXt architectures while maintaining or improving performance across various classification tasks.

- Through salience analysis, we demonstrate clear evidence of semantic reasoning by revealing that intermediate spatial layers consistently focus on meaningful object parts, which SVC encodes using data-driven, psychovisual coding–inspired filters. Abstractions produced by SVC are shown to form structured encodings of the object parts, whose use in final decision making mirrors human psychovisual processing and highlights a promising pathway toward interpretable model reasoning.

## 2 BACKGROUND

In this section, we review related prior computer vision works on methods with biological motivations, as well as those that use the frequency domain. Additional details/background about psychovisual coding, the Fourier Transform and complex-valued neural networks are provided in Appendix A.

**Biologically Inspired Vision.** Biologically inspired approaches in computer vision predominantly focus on modelling early vision stages. In particular, much attention has been given to receptive fields (RF) - regions of visual stimuli that elicit strong neural responses in the visual cortex. Mammalian RFs are known to act as directional differential operators, closely resembling traditional image processing functions like wavelets and Gabor filters (Olshausen & Field, 1996; Hubel & Wiesel, 1962; Ringach, 2002). These parallels motivated their use in approximating low-level human vision, serving as effective feature extractors for basic visual structures like edges and shapes. In deep learning, these functions have been used to build neural networks that mimic cortical pathways (Liu et al., 2023), and early-layer CNN kernels also perform similar directional operations (Krizhevsky et al., 2017; Rippel et al., 2015). Beyond RFs, cortical responses have also been modelled from a frequency domain perspective.

Research conducted in the 1990s by French researchers led by Dominique Barba in understanding human vision included psychovisual experiments determining frequency sensitivities of the human visual cortex (Saadane et al., 1994; Senane et al., 1995; Saadane et al., 1998). These informed the design of psychovisual coding, an image quantization and compression scheme that is perceptually lossless to humans. It first decomposes the frequency domain into a number of coronal sub-bands (Figure 2 (left)), then for each sub-band applies specific quantization thresholds derived based on the discovered sensitives. This enabled only frequencies corresponding to perceptually salient image figures to be encoded in full, while the rest are removed or heavily compressed without affecting perceived image quality. An extended review covering the motivations and additional background for psychovisual coding is included in Appendix A.1.

**Frequency Domain Learning.** Frequency analysis has long been a staple in traditional image processing. Unlike the spatial domain, which is highly localised and expresses features in contiguous pixel neighbourhoods, the frequency domain is more conducive to global representations (see Appendix A.2). Formulated in this space, image processing functions like ridgelets (Candés & Donoho, 1999), curvelets (Starck et al., 2002) and contourlets (Do & Vetterli, 2005) have appealing sparse representations. In fact, they bear a strong resemblance to psychovisual codes since they target specific selections of sub-bands, corresponding to features from different spatial scales. Although these functions have been incorporated into neural networks before, they are only effective on small problems due to their handcrafted nature (Liu et al., 2021).

In deep learning, the frequency domain has primarily been used to exploit the Convolution Theorem (Gonzalez & Woods, 2014), whereby spatial circular convolution becomes simple elementwise multiplication in the frequency domain. Many works leveraging this property are performance-driven: (Li et al., 2020a; Chi et al., 2020; Guan et al., 2021) use frequency-domain filters to accelerate CNNs and incorporate global context, while (Rao et al., 2023; Lee-Thorp et al., 2021; Huang et al., 2023) employ global frequency filters as lightweight and effective token mixers for transformer-style models. Other studies use frequency-domain re-parameterizations of CNNs to analyse model properties and behaviours (Rippel et al., 2015; Grabinski et al., 2023; Kabri et al., 2023), such as optimal kernel structures. In contrast, our work contributes to frequency-domain representation learning, a direction that remains comparatively underexplored. Since global context is crucial for high-level features (Rao et al., 2023; Dosovitskiy et al., 2021), the frequency domain provides a natural setting in which to represent and process semantic structure. Our SVC module employs learnable band-limited frequency filters—data-driven analogues of hand-crafted visual codes—to encode task-relevant semantic information. Unlike prior works that integrate frequency filters directly into their base computational blocks, SVC acts as an abstraction layer bridging feature extraction and decision. Another related work uses frequency filters to select and amplify domain-transferable frequency (Lin et al., 2023), similar to how SVC selects task-relevant features, but their approach is not framed as representation learning nor is it used for interpretability. Moreover, SVC also enables the novel interpretable psychovisual-like processing achieved by PsychoNet.

## 3 METHOD

**Semantic Visual Coding.** A $N \times N$ digital image, or a spatial feature map derived from it by a neural network, can be viewed as a 2D discrete signal $x[m, n], \; m, n \in 0, ..., N$. This can be represented in the frequency domain as a linear combination of complex-valued sinusoids via the

2D DFT(Cooley et al., 1969):

$$X[u,v] = \frac{1}{N^2} \sum_{m=0}^{N-1} \sum_{n=0}^{N-1} x[m,n] e^{-2\pi i \left( \frac{um+vn}{N} \right)} \quad (1)$$

where $i$ denotes the imaginary unit. These weights are the (frequency) spectrum of the image and is a complex-valued space known as the frequency domain, which can be computed efficiently using the Fast Fourier Transform (FFT) (Cooley et al., 1969). Psychovisual coding (Saadane et al., 1998) partitions this space into radial sub-bands (2 (left)), and assigns each a threshold corresponding to sensitivity to human vision. These thresholds decide the level of granularity when quantizing images, so that perceptually important features are preserved while others are coarsely represented or discarded.

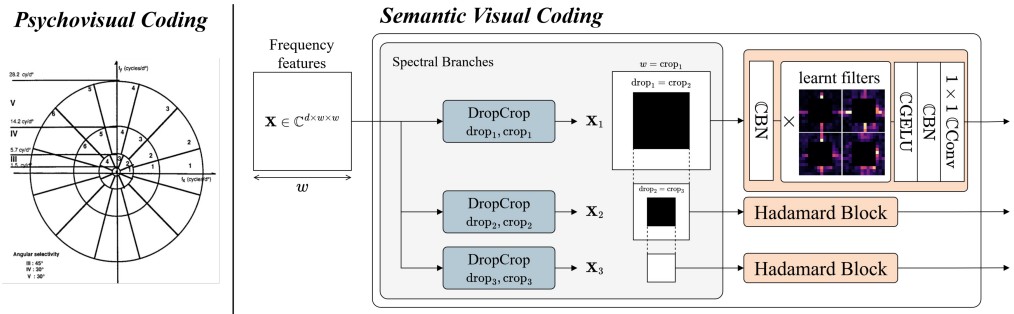

Figure 2: **(Left) Hand-crafted psychovisual coding** from Saadane et al. (1998), which quantizes perceptually salient radial frequencies determined by human vision experiments. **(Right) Our Semantic Visual Coding module,** a data-driven adaptation of psychovisual coding. It uses (1) *Spectral Branches* for radial spectral decomposition (2) *Hadamard Blocks* to apply learnt element-wise filters and channel mixing. $\mathbb{C}$Conv/BN/GELU denote complex-valued convolution, batch norm and GELU operations - see Appendix B.2

We introduce *Semantic Visual Coding* (Figure 2 (right)) which aims to generalize this coding principle beyond low-level vision and adapt it to high-level features in deep network layers. In this setting, the selection of frequencies should no longer be fixed by handcrafted thresholds, but instead learnt directly from data to encode task-relevant semantic information. Semantic Visual Coding has the following formulation:

Let $\boldsymbol{X} \in \mathbb{C}^{d \times w \times w}$ be frequency domain input features, where $d$ is the number of channels and $w \times w$ the spatial size.

1. We apply *Spectral Branches* which replicate the radial frequency partitioning in psychovisual codes. These divide $\boldsymbol{X}$ into disjoint rectangular sub-bands $\boldsymbol{X}_1, \boldsymbol{X}_2, ...$ using *DropCrop* blocks, which set a lower frequency boundary ($\text{drop}_i$) by zeroing central frequencies and an upper boundary ($\text{crop}_i$) by cropping $\boldsymbol{X}$ to size $d \times \text{crop}_i \times \text{crop}_i$.

2. For each sub-band $\boldsymbol{X}_i$, *Hadamard Blocks* apply a set of learnt filters $\boldsymbol{W}_i \in \mathbb{C}^{d \times \text{crop}_i \times \text{crop}_i}$ via element-wise (Hadamard) multiplication. Additionally, we also apply Softmax across the channels of $\boldsymbol{W}_i$ to amplify important frequency selections and suppress unimportant ones, emulating the quantization in psychovisual coding.

3. Hadamard Blocks further apply complex $1 \times 1$ convolution block to mix together different information extracted by each channel/filter, yielding our final representations.

In practice, for all models we apply Spectral Branches at a spatial resolution of $w = 14$ and use three sub-bands with $[\text{crop}_i, \text{drop}_i]$ values of $[14, 8]$, $[8, 4]$ and $[4, 1]$ respectively. More details can be found in Appendix B.

**PsychoNet.** The PsychoNet framework (Figure 3) adapts standard spatial CNNs to use Semantic Visual Coding. This setting enables experimentation to assess if our codes produce meaningful abstract representations that support interpretable semantic reasoning, as well as practical performance evaluation against standard baselines.

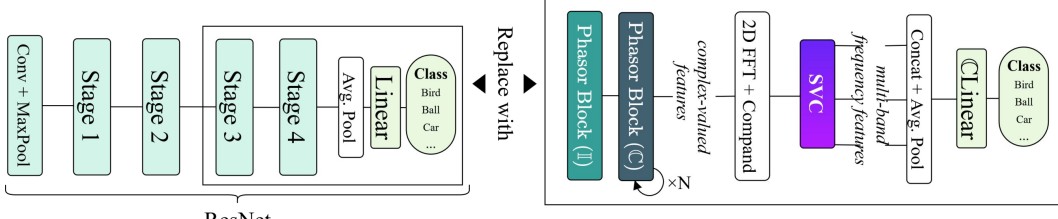

Figure 3: Example: Converting ResNet50/101 image classification models with PsychoNet.

In our experiments, we apply PsychoNet to ResNet (He et al., 2016) and ConvNeXt (Liu et al., 2022) architectures. Below we summarise the main steps for implementing PsychoNet below while full architectural configurations are provided in Appendix B.

1. A number of low-level feature extraction layers are retained from the base CNN, for example, the first two resolution stages in the case of ResNet50 and ResNet101.

2. The remaining spatial layers are replaced with Phasor Blocks, described further below. Compared to the original CNNs layers, Phasor Blocks typically use higher spatial resolution and only downsample down to $14 \times 14$ instead of $7 \times 7$. Though this increases FLOPs (Appendix C), we found there is insufficient granularity at $7 \times 7$ to clearly separate low and high frequencies after FFT.

3. 2D FFT is applied to convert features from the spatial to frequency domain. As with most visual features, the magnitude of $X$'s DC (0 frequency) and low frequency features typically dominate over those of high frequency ones, so we use a simple companding operation to reduce this imbalance (Appendix B).

4. SVC is applied, and its outputs from each frequency band aggregated. These are then used for output prediction directly in the frequency domain using a complex-valued linear layer.

The ConvNeXt-based PsychoNet differs from its ResNet-based counterpart in only two key architectural aspects: (1) the use of ConvNeXt's $4 \times 4$ patch embedding layer at the input, and (2) $7 \times 7$ depthwise convolution blocks are used instead of regular convolution layers within Phasor Blocks, matching ConvNeXt's primary computational block. Despite their minimal nature, these adaptations are sufficient to recover performance close to ConvNeXt, as demonstrated in Section 4.

The following section outlines the motivation and formulation of Phasor Blocks, with additional architectural details provided in Appendix B

**Phasor Blocks.** Filtering in the frequency domain is powerful as it captures information encoded as both magnitude and phase. However, PsychoNet operates on real-valued inputs (natural images), and real features incur the conjugate symmetry of the Fourier Transform (FT), rendering half of the frequency domain redundant. While this constraint matters little when the frequency domain is used solely for convolution (Rao et al., 2023; Li et al., 2020a), it limits learnt filtering from fully exploiting complex representations. To address this, we introduce Phasor Blocks (Figure 4) to augment real-valued spatial features with complementary complex-valued ones, breaking conjugate symmetry and improving the specificity of learned sub-bands. In practice, imaginary components are generated from existing real features using lightweight depthwise convolution based blocks which decouple spatial and channel mixing to encourage cross-channel interaction without altering spatial structure. In natural complex signals, real and imaginary components convey complementary information at the same spatial location (Gonzalez & Woods, 2014; Lee et al., 2022), making it important that the generated imaginary features do not introduce substantial new spatial information. As demonstrated in later sections, this design enables Phasor Blocks to extract meaningful object parts that form the basis of SVC abstractions supporting interpretable semantic reasoning. Further evidence from the ablations in Appendix D.1 and the salience maps in Figure A.10 shows that while SVC alone encourages attention to object parts, the inclusion of Phasor Blocks and their complex representations leads to substantially clearer and more localised part-based salience, enabling greater interpretability.

Figure 4 provides an overview of the two Phasor Block configurations used in this work, with full architectural diagrams and design details presented in Appendix B.3.

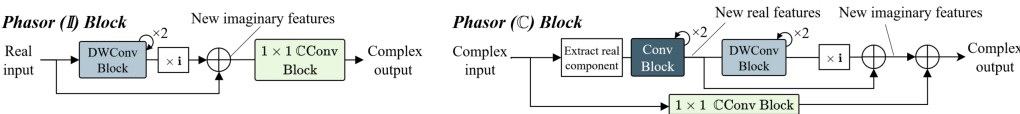

Figure 4: **Phasor Blocks architectures**. Phasor ($\mathbb{I}$) blocks generate complementary imaginary features for real-valued input. Phasor ($\mathbb{C}$) blocks generate additional complex features for complex-valued input based on its real component. Implementations of normal convolution (Conv), depthwise convolution (DWConv) (Liu et al., 2022) and complex-valued convolution ($\mathbb{C}$Conv) blocks for each of our models are presented in Figure A.2

## 4 EXPERIMENTS AND RESULTS

We applied PsychoNet across multiple ResNet architectures, which provide straightforward and well understood baselines, as well as ConvNeXt-S, as an example of a state-of-the-art CNN. Quantitative evaluation was conducted on image classification across small to large-scale datasets: CIFAR-10/100 (Krizhevsky, 2009) both contain ~50K low-resolutions images, while ImageNet-100 is a moderate sized subset (~130K images) of ImageNet (Deng et al., 2009). Finally, we also use the standard large ImageNet-1K subset containing ~1.2 million training and ~50K validation images. Full dataset, training and hardware details are presented in Appendix C, and full model configurations in Appendix B.

| Model | Param. (M) | # Layers | CIFAR-10 | CIFAR-100 | IN100 | IN1K |
|---|---|---|---|---|---|---|
| ResNet50 | 25.56 | 54 | 94.14 | 78.10 | 80.90 | 76.04 |
| Psycho-S | 25.35 | 65 ↑ *20.4% more* | **95.08** | **78.97** | **82.50** | **76.86** |
| ResNet101 | 44.55 | 105 | 93.64 | 79.13 | 81.90 | 78.43 |
| Psycho-B | 42.01 | 93 ↓ *11.4% less* | **94.99** | **79.49** | **83.60** | **78.85** |
| ResNet152 | 60.10 | 156 | 93.17 | 77.51 | 83.60 | 79.59 |
| Psycho-L | 61.28 | 93 ↓ *40.4% less* | **94.95** | **79.64** | **84.82** | **79.85** |
| ResNet270 | 89.60 | 276 | 76.51 | 50.87 | 83.80 | 80.01 |
| Psycho-H | 88.61 | 93 ↓ *66.3% less* | **94.68** | **79.89** | **85.00** | **80.45** |
| ConvNeXt-S | 50.22 | 113 | 94.09 | 76.96 | **86.98** | **80.78** |
| PsychoDW | 49.51 | 106 ↓ *6.2% less* | **95.46** | **79.67** | 86.76 | 80.59 |

Table 1: Summary of classification results (% top-1 accuracies) on CIFAR-10, CIFAR-100, ImageNet-100 (IN100) and ImageNet-1K (IN1K). Each pair of rows (separated by horizontal lines) compares a baseline CNN and the PsychoNet based on it. Further detailed results are presented in Appendix C.

A common characteristic of newer CNNs like ConvNeXt is their reduced dependence on depth for model scaling (Liu et al., 2022; Xie et al., 2017), whereas large ResNet architectures heavily add additional high-level layers to increase representational capacity (He et al., 2016). We hypothesize that since high-level processing in PsychoNet is handled by our frequency domain modules, it should also be much less depth-dependent than ResNet. The Psycho-S/B/L/H models, counterparts to ResNet50/101/152/270, were designed to test this and do not increase Phasor Block depth beyond the Psycho-B/ResNet101 size. Instead, parameter parity is maintained by widening (increasing number of feature channels) the existing Phasor Blocks and SVC filters—an equally simple, if not simpler, scaling strategy than the depth expansion used in ResNet. Table A.2 compares the resulting channel-width configurations across these model sizes.

Overall, we found that PsychoNet slightly improves the performance of each baseline ResNet across all datasets, despite Psycho-L and Psycho-H using ~1.7× and ~3× less layers than their ResNet

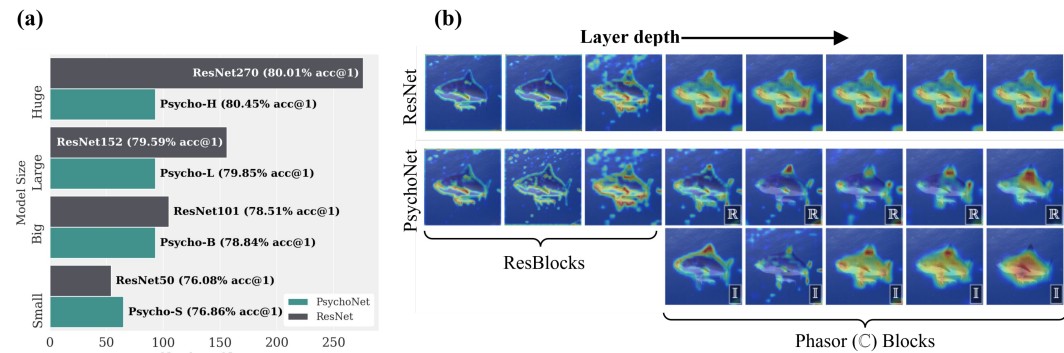

Figure 5: **(a)** Comparison of model depth when scaling ResNet vs. PsychoNet. **(b)** Comparison between activation maps (via KPCA-CAM) of Psycho-B and ResNet101 for a range of layer depths. Real and imaginary components are denoted by $\mathbb{R}$ and $\mathbb{I}$.

baselines respectively (Table 1, Figure 5 (a)). This demonstrates that SVC is able subsume the role of a significant portion of deep spatial layers, showing it is an effective high-level processing module. We believe this arises because SVC operates on globally focused frequency-domain representations, in contrast to spatial features which are sparse and locally structured, requiring significant depth to progressively aggregate features. We also note that ResNet152 and ResNet270 underperform smaller ResNets on CIFAR-10/100, likely as their large depth is unsuited for the low-resolution images in these datasets. In comparison, our much shallower Psycho-L/H models were unaffected.

On ConvNeXt-S, PsychoDW achieves comparable performance despite its Phasor Block being only a simple adaptation of the ResNet-oriented one, indicating that further targeted adaptations may yield even stronger performance. This demonstrates that PsychoNet extends naturally to modern CNN architectures beyond ResNet. As shown in the following section, PsychoNet also adds interpretable, psychovisual-style reasoning to both ResNet- and ConvNeXt-based models while maintaining performance, which will be particularly beneficial for future applications such as medical image analysis that require transparent and trustworthy models.

Finally, we also acknowledge that our PsychoNet models use considerably more FLOPs than their respective CNN baselines (Table A.12). The increased computation is attributed to (1) Phasor Blocks requiring higher-resolution features than the CNN layers, to support $14 \times 14$ SVC filters and (2) complex-valued operations (complex convolution etc.) being poorly optimised in deep learning frameworks. Regarding (1), Appendix D.1.1 presents FLOP-efficient variants of Psycho-S/B that adds an early downsampling step to $7 \times 7$ and extracts the $[14, 8]$ sub-bands from intermediate Phasor Block features. On ImageNet-100, this preserves Psycho-S/B performance while matching the FLOPs of ResNet-50/101. While these findings suggest that efficient PsychoNet variants are achievable, our current focus remains on establishing the first instance of interpretable psychovisual separation of low and high-level processing in CNNs, with supporting results presented in the following section.

**Filter learning.** Figure 6 visualises SVC filters learnt by ResNet-based PsychoNet, showing the top spatial principal components as an approximation of the most important frequency features. We find that filters across every sub-band learn very sparse selections of frequencies; similar results were found for the ConvNeXt-S based PsychoDW and are shown in Figure A.5 (a). However, this requires sufficiently large training corpora - the ImageNet-100-trained filters are noticeably noisier than for ImageNet-1K, and those for CIFAR-10/100 (Figure A.4) even more still. This suggests that these patterns correspond to a data-driven representation naturally emergent from visual information. Additionally, we ran ablation experiments to evaluate the effects of Phasor Blocks and Spectral Branching (Appendix D.1). Removing Phasor Blocks (and replacing them with ResNet-style residual bottleneck blocks (ResBlocks)) removes complex-valued spatial features and introduces conjugate symmetry to the frequency domain. This yields symmetric filter features that are far less expressive. Likewise, removing the spectral decomposition of Spectral Branches (we use one global

filter instead of three band-limited ones) also reduces filter sparsity. This is likely as exposure to the entire frequency domain makes it harder for filters to specialise to specific sub-bands.

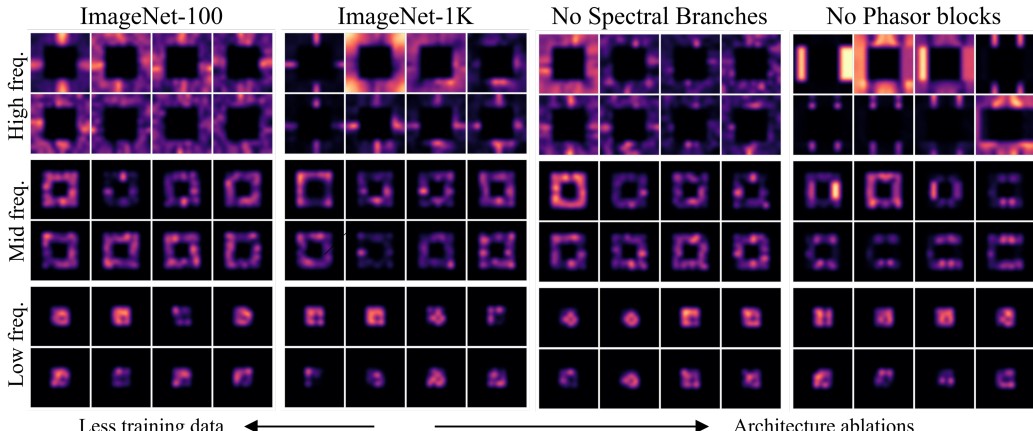

Figure 6: SVC filters learnt by Psycho-B trained on ImageNet-100 and ImageNet-1K, as well as for two ablation models on ImageNet-1K. Bilinear smoothing has been applied. 'High/mid/low freq.' refer to the [14, 8], [8, 4] and [4, 1] frequency sub-bands created by Spectral Branches. 'No Spectral Branches' removes Spectral Branches and uses a single Hadamard Block with global filters - we extract sub-bands only for the visualisation. 'No Phasor Blocks' replaces all Phasor Blocks with ResBlocks.

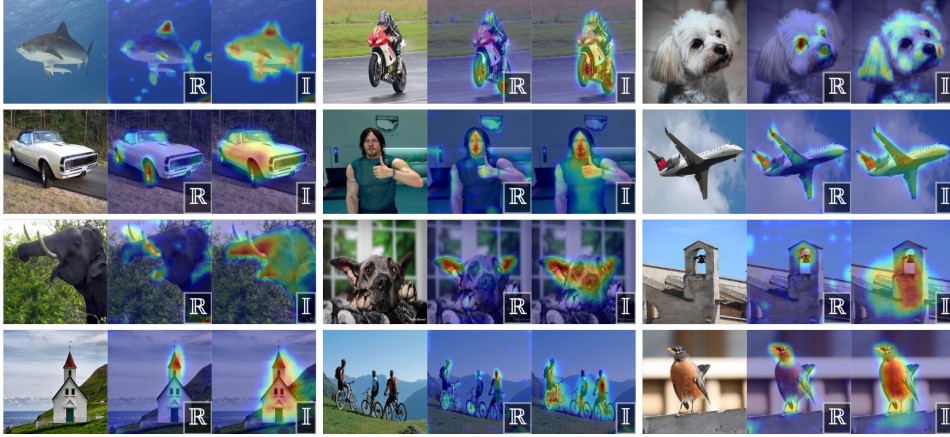

Figure 7: Assorted activation maps (via KPCA-CAM) for mid-level Phasor Blocks of Psycho-B. Real and imaginary components are denoted by $\mathbb{R}$ and $\mathbb{I}$. An equivalent visualisation for PsychoDW is shown in Figure A.6.

**Representation Analysis.** As our quantitative results suggested SVC likely subsumes the high-level processing of deep ResNet layers, we visualise layer activations using KPCA-CAM (Karmani et al., 2024) to compare spatial processing between the two models (Figure 5 (b)). This approach generates salience maps by projecting activations onto the first principal component of their kernel PCA. ResNet's early layers target low-level features (edges), but later salience regions quickly grow to cover the entire subject and it is not particularly clear on which parts of the shark each layer is focusing. This likely reflects ResNet's use of homogenous layer stacks, which, without explicit structure, may learn layers that perform diffuse and weakly organised operations. In contrast, early-mid level Phasor Blocks clearly fixate on morphological features of the shark, such as its snout, fins and tail. Figure 7 shows further examples of Phasor Blocks localising key characteristics of different object categories, such as dog ears, elephant tusks and car wheels. Similar results visualising activation maps of PsychoDW, and comparing them to those of ConvNeXt-S, are presented

in Figures A.5 (b) and A.6. Since KPCA-CAM only uses activations of the visualised layer and is uninfluenced by model predictions (e.g. via backpropagation in gradient-based CAMs), these results indicate that Phasor Blocks specalize to extract meaningful semantic object parts. This organisation is likely shaped by the presence of SVC downstream, which we show below encodes these parts into higher-level abstractions that support interpretable semantic reasoning. This is further support by the salience maps in Figure A.10 that shows although SVC alone does promote object-part focus, clear part-level isolation only emerges with the addition of Phasor Blocks and their complex features.

Interestingly, it appears that the imaginary components of Phasor Block activations capture more global features than the real components (i.e. a dog's face vs. its ears). An initial clustering visualisation of Phasor Block activations is also presented in Figure A.8, which finds that observable clustering emerges both components which becomes increasingly pronounced at deeper layers. These findings suggest that Phasor Blocks learn a rich utilisation of complex-valued representations that is iteratively refined through each block.

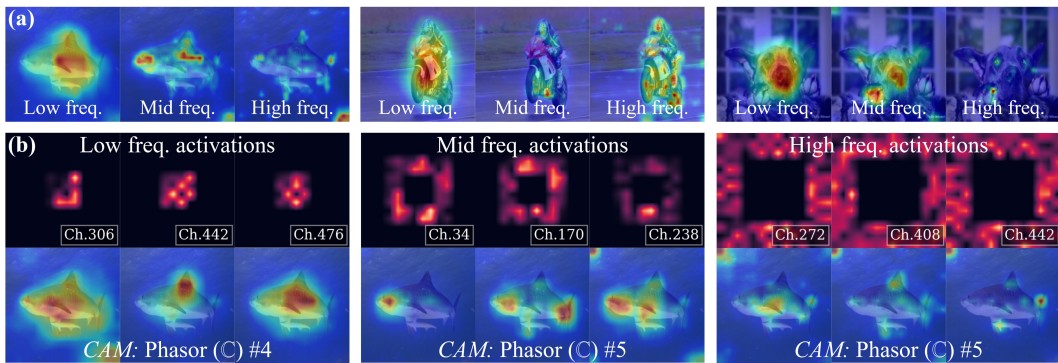

Figure 8: Psycho-B Phasor Block salience maps (via HiResCAM) conditioned on gradients **(a)** from individual Spectral Branch sub-bands and **(b)** from individual frequency domain feature channels.

For initial exploration of SVC's encoding mechanisms, we use HiResCAM (Draelos & Carin, 2020) for gradient-based activation visualisation. It produces salience maps by element-wise multiplying layer activations with gradients backpropagated from model predictions, so in classification the salience regions have a high contribution to the class prediction. We extend this approach to isolate regions used by specific parts of SVC by first masking (setting to zero) gradients from the other components, enabling exploration of how SVC encodes Phasor Blocks features. First, we examine each of the three sub-bands created by PsychoNet's Spectral Branches. After masking gradients of Hadamard Blocks for all but one of the sub-bands, Phasor Blocks' salience regions reveal that SVC distributes object parts by scale. Figure 8 (a) shows that the low-frequency sub-band focuses on subjects broadly, while mid-high frequencies isolate more specific parts of different sizes. This aligns with frequency domain theory, in which low frequencies capture coarse spatial structure and higher frequencies finer detail and edges, supporting the view that SVC performs structured filtering in the frequency domain. We also isolate activations from individual Hadamard Block channels, showing that within each band, channels specialise to distinct object parts and correspond to distinct sparse frequency selections (Figure 8 (b)). This analysis was also applied to PsychoDW with similar results, shown in Figure A.7.

Overall, these result suggest that SVC learns a semantic intermediate representations that encodes selections of object parts. Given that SVC is placed immediately before the decision making (classification) layers of PsychoNet, it is likely selecting those most relevant to the task. In doing so, SVC functions as an abstraction bridging part extraction in Phasor Blocks and higher-level semantic reasoning, mirroring the role of abstractions used in psychovisual processing.

**Limitations and Future Work.**  A key limitation of our work is that, although we show SVC organises and encodes selections of meaningful object parts, future research is still required to determine how the deeper semantic meaning of these abstractions should be interpreted in relation to broader notions of reasoning, such as those studied in neuroscience (Quiroga et al., 2005; Kriegeskorte et al., 2008). We will also explore addressing the high FLOP usage of PsychoNet by exploring

optimisations for complex-valued operations (e.g. employing Cauchy-Riemann identities (Ahlfors, 1979)) as well as more sophisticated formulations of Phasor Blocks.

Additionally, it would also be insightful to explore applying PsychoNet to broader task types, particularly image-to-image tasks like segmentation which may allow SVC to utilize wider frequency ranges than classification. Another focus will be practical applications in MRI, to explore how SVC's semantic abstractions can enhance transparency and trustworthiness in models operating directly on k-space data. Finally, it is also known that aliasing can afflict standard CNN architectures (Grabinski et al., 2022); future work should assess its impact on our frequency-domain representations and whether mitigation can improve results.

## 5 CONCLUSION

In this work, we introduced Semantic Visual Coding (SVC), the first high-level vision representation learnt in the frequency domain that produces sparse, data-driven coronal selections of discrete Fourier space. Our PsychoNet framework integrating SVCs show that it can maintain performance across multiple classification datasets, but is less depth-dependent, suggesting that SVC improves high-level processing previously done by deep spatial layers. In contrast to the unorganised processing of conventional CNNs, we find that PsychoNet clearly separates processing stages: Phasor Blocks extract semantically meaningful object parts, while SVCs encode and organise these parts into sparse, frequency domain representations used to make classification decisions that can be visualized. This pipeline provides strong evidence that it may mimic intermediate abstractions used by the brain to separate feature extraction from higher cognition as suggested in previous neuroscience studies. While further work is required to understand the reasoning mechanisms of SVCs, it is clear that frequency domain abstraction is a promising direction for interpretable human-like model reasoning.

## ACKNOWLEDGEMENTS

In accordance to ICLR 2026 guidelines, we acknowledge the use of large language models (LLMs) in preparing this manuscript. Their role was limited to assisting with editing and polishing writing.

Additional acknowledgements will be added after deanonymization.

## ETHICS STATEMENT

All authors have reviewed the ICLR 2026 code of ethics and verified to the best of our knowledge that the work in our paper conforms with it. In particular, this work introduces a new theoretical framework, so it is unlikely to cause direct harm or negative impacts to society. Additionally, we only use open datasets such as ImageNet, so privacy concerns do not arise, though we acknowledge that these datasets contain known biases that may influence model behaviour.

## REPRODUCIBILITY STATEMENT

We are committed to ensuring the reproducibility of our results. All experiments in this work were conducted on publicly available datasets, which have been appropriately cited. Detailed training recipes and hardware details are presented in Appendix C and Appendix D.1. Detailed model configurations are presented in Appendix B. After deanonymization, we will also release our code repository including training scripts, model weights and instructions to reproduce all of our results.

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

# Appendices

In the following we present appendices to our work, structured as follows: Appendix A presents additional background material. Appendix B provides detailed information about the architectural configurations of all models used in our experiments. Appendix C presents results, dataset information and training recipes for all of our classification experiments. Appendix D.1 provides full results and details for PsychoNet architectural ablation studies.

## A  BACKGROUND

In this section we present additional background and details about psychovisual coding, the Fourier Transform and complex-valued networks.

### A.1  PSYCHOVISUAL CODING

Our work is inspired by groundbreaking research conducted in the 1990s by French researchers led by Dominique Barba in understanding the human aspect of mammalian vision, i.e. the psychovisual capability of the human brain for visual perception arising from the need for early television signal compression (Hanen & Barba, 1993). At the time, statistical approaches based on Shannon's information theory and rectilinear methods such as discrete (Haar) wavelets were popular, and they argued that these approaches were sub-optimal because they treated all errors equally. They would propose psychovisual quantizers as an efficient form of image coding that would retain the important image information pertaining to its interpretation by the human vision system and quantization matched the detection thresholds of the visual cortex (Senane et al., 1995). These quantizers were proposed to be the coronas of the 2D Fourier space, where the model of the vision system assumes Fourier space is analyzed using radial symmetric functions (Saadane et al., 1994; 1998) (see Figure A.1), which they showed can also be mapped to colors in human vision (Callet et al., 1999). The premise is that visual recognition and feature extraction could be performed by selecting coronal sectors of Fourier space directly through the quantisation of adjacent frequencies, thereby providing directional band limited filtering within the scene. Their psychophysical experiments were also used to select the optimal sub-bands that allowed image compression that was difficult for humans to distinguish (Saadane et al., 2001). Our SVC is a data-driven adaptation of this approach, using band-limited frequency filters to learn sparse frequency selections using supervisory signals from a classification task.

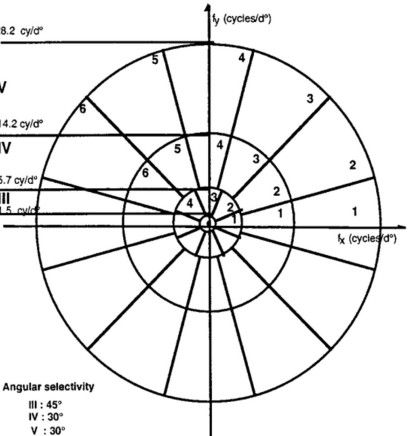

Figure A.1: Coronal frequency sub-bands used in psychovisual coding from Saadane et al. (1998).

This work on visual codes over the course of a decade would result in among the first uses of vector quantization for image coding (Senane et al., 1995), a perceptually based image quality metric (Saadane et al., 2001) and one of the foundations of discrete projection theory, where a central

slice theorem is established for discrete Fourier space based as exact 1D forms of these psychovisual radial functions as slices and therefore projections in image space (Guedon et al., 1995). This work would even pioneer the use of the wavelet transform to projection data before it would be formalized as ridgelets by Candes and Donoho (Candés & Donoho, 1999). The Mojette transform would itself form the basis of an entire area of discrete tomography that creates discrete projections of images (Normand et al., 1996) in diverse areas such as image reconstruction (Kingston & Svalbe, 2007; Chandra et al., 2014) and compression (Guédon et al., 2001), computed tomography (Hou & Zhang, 2013) and network transmission (Parrein et al., 2001; Verbert et al., 2002). Although the number of publications is too numerous to list here, a summary of these works and areas can be found in the Mojette transform book (Guédon, 2013).

## A.2 THE FOURIER TRANSFORM AND THE FREQUENCY DOMAIN

In Section 3 we only describe the 2D DFT as digital images are discrete 2D signals in the spatial domain, while the standard FT operates on continuous signals. The DFT is derived by first viewing a discrete signal as the product of a continuous signal and a sequence of unit impulses (sampling), applying the FT to yield a continuous function in the frequency domain, then sampling it again to discretize it. Detailed derivations of both the FT and DFT may be found in most image processing texts, such as (Gonzalez & Woods, 2014). There are also inverse transforms, namely the Inverse Fourier Transform (IFT) and Inverse Discrete Fourier Transform (IDFT), for transforming frequency domain signals back into the spatial domain. While we do not use them in PsychoNet, they reflect the duality between the spatial and frequency domains - any operation in one domain has a counterpart in the other. The most famous example of this relationship is the Convolution Theorem.

Let $x[u, v], y[u, v], u, v \in 0, ..., N - 1$ be two discrete $N \times N$ spatial signals. The circular convolution of these two signals is defined as:

$$x[u, v] * y[u, v] = \frac{1}{N^2} \sum_{m=0}^{N-1} \sum_{n=0}^{N-1} x[m, n] y[((u - m))_N, ((v - n))_N] \tag{2}$$

where $(.)_N$ denotes modulo $N$. The Convolution Theorem (Gonzalez & Woods, 2014) then states that:

$$\mathcal{F}[x * y] = \mathcal{F}[x] \odot \mathcal{F}[y] \text{ or equivalently } x * y = \mathcal{F}^{-1}[\mathcal{F}[x] \odot \mathcal{F}[y]] \tag{3}$$

where $\mathcal{F}[.]$ and $\mathcal{F}^{-1}[.]$ denote the DFT and IDFT, and $\odot$ the Hadamard product. Hence, circular convolution in the spatial domain is equivalent to applying the Hadamard product in the frequency domain. As such, the frequency domain is highly conducive to global representations, since each element of an image's frequency spectra presents a unique global view of the image, analogous to convolving it with a directional striped kernel.

In practice, the DFT and IDFT are computed using the Fast Fourier Transform and Inverse Fast Fourier Transform respectively (Cooley et al., 1969). Note that if $x$ and $y$ were multi-channel features instead, i.e. of dimension $d \times N \times N$ for $d$ channels like the input features and learnt filters of our Hadamard Blocks, then the frequency domain Hadamard product is equivalent to circular *depthwise* convolution in the spatial domain. Unlike the Conv2D operation of CNNs, this does not mix channels, which is why both of our Hadamard Blocks and the Global Filter block from Rao et al. (2023) include explicit channel-mixing via $1 \times 1$ convolution layers.

## A.3 COMPLEX-VALUED NEURAL NETWORKS

Most work for complex-valued neural networks involve developing components of these networks to work in the complex domain, such as activation functions (Scardapane et al., 2018). Most complex-valued CNNs use the network blocks introduced by Trabelsi et al. (2018). The distributive property of convolution allows convolution between a complex input $\boldsymbol{h} = \boldsymbol{a} + i\boldsymbol{b}$ and a complex kernel $\boldsymbol{W} = \boldsymbol{W_R} + i\boldsymbol{W_I}$ to be decomposed into four real-valued component wise convolutions:

$$\boldsymbol{W} * \boldsymbol{h} = (\boldsymbol{W_R} * \boldsymbol{a} - \boldsymbol{W_I} * \boldsymbol{b}) + i(\boldsymbol{W_I} * \boldsymbol{a} + \boldsymbol{W_R} * \boldsymbol{b}) \tag{4}$$

Consequently, complex-valued convolution layers are usually more computationally and memory intensive (additionally stores imaginary features) than real-valued ones. Trabelsi et al. (2018) also

developed complex normalization methods and activation functions. Complex-valued modules in PsychoNet use the complex-valued convolution (ℂConv) and batch-normalization (ℂBN) layers from Trabelsi et al. (2018), and a naïve adaptation of the GELU activation function (ℂGELU) which just applies the original function to real and imaginary channels separately.

When applying complex-valued networks to real-valued images, most works use a small initial module to convert the input into complex-valued features. However, such approaches have yielded only minor improvements in the past over directly using real-valued networks (Trabelsi et al., 2018; Li et al., 2020b). Accordingly, recent complex-valued networks predominantly focus on domains with naturally complex data, such as MRI, radar and audio signal processing (Dedmari et al., 2018; Vasudeva et al., 2022; Cole et al., 2021; Lee et al., 2022; Trabelsi et al., 2018). To try bridge this gap, a complex-valued colour space by reinterpreting the cylindrical coordinates of the HSV colour model as 2D magnitude and phase was developed (Yadav & Jerripothula, 2023). They applied this to standard complex-valued CNNs, improving results on common image classification tasks, but retained the high complexity of complex-valued networks. On the other hand, PsychoNet primarily uses real-valued modules that learn to generate *complementary* complex-valued features to given real features, as described in Section 3.

## B    MODEL CONFIGURATIONS

Here we provide full details of the architectural configurations of all of our models. For all tables, we use the ResNet approach of counting the number of model layers as the number of convolutional and linear layers; each element-wise filter block in Hadamard Blocks are also counted as one layer.

### B.1    CNN BASELINES

The ResNet50, 101 and 152 models we use are from He et al. (2016) and are implemented in most common deep learning frameworks (we use the one from `PyTorch` (Paszke et al., 2017)). For ResNet270, we follow the block configurations in Bello et al. (2021), but do not implement any of the newer blocks/layers they also introduce, so it purely just adds more residual bottleneck blocks (ResBlocks) to ResNet152 for fair scaling. Table A.1 compares the sizes of the four ResNet models as well as their block configurations, grouped by feature resolution (which are $56 \times 56$, $28 \times 28$, $14 \times 14$ and $7 \times 7$).

Table A.1: ResNet block configurations.

| Model | Parameters (M) | # Layers | # Blocks |
|---|---|---|---|
| ResNet50 | 25.56 | 54 | `[3-4-6-3]` |
| ResNet101 | 44.55 | 105 | `[3-4-23-3]` |
| ResNet152 | 60.19 | 156 | `[3-8-36-3]` |
| ResNet270 | 89.60 | 276 | `[4-29-53-3]` |

For ConvNeXt-S, we follow the original implemention in Liu et al. (2022).

### B.2    PSYCHONET

Figure A.2 summarises the key configuration details of each PsychoNet variant, namely the feature resolution and channel width at each Phasor Block and the number of filters used in the SVC module. Further architectural details for each model are reported in Tables A.3 through A.7. For Phasor Blocks, we list each layer using the '**resolution**: layer configuration' format. The ResNet-based PsychoNet models use the same initial input embedding layer as ResNet ($7 \times 7$ Conv2D and maxpooling) is used, while PsychoDW uses the same $4 \times 4$ patch embeddings as ConvNeXt-S. Interestingly, we found that using the initial layers of ResNet50, instead of ConvNeXt-S, in our ConvNeXt-S based PsychoDW actually yielded better results (approx. $\uparrow 0.5\%$ top-1 accuracy on ImageNet-1K), so we chose to use it for the model. However, we do change all ConvBlocks in Phasor ($\mathbb{C}$) (see Figure A.2) to depthwise convolution blocks to maintain general faithfulness to the ConvNeXt model.

Finally, the companding operation we apply after taking the 2D FFT (in Figure 3) simply zeros the DC component and applies the element-wise function:

$$x \in \mathbb{C}, \quad \text{Compand} : x \to |x|^{\frac{1}{1.25}} \cdot \exp(i\angle x) \tag{5}$$

where $|x|$ denotes the magnitude of $x$ and $\angle x$ its phase. Since the exponent applied to the magnitude is $\in (0, 1)$, this function compresses frequencies of large magnitude (i.e. frequencies very close to the DC component), and expands the magnitude of those further from it.

Table A.2: Configuration summary of different PsychoNet variants. For Phasor Blocks, we display **resolution**: [#channels per block], and '$(\mathbb{I})$' denotes a Phasor Block $(\mathbb{I})$. #Filters denotes the number of channels of element-wise filters per sub-band of SVC. #Layers show overall layers / complex convolution layer counts.

| Model | Phasor Blocks | #Filters | #Layers | Params (M) |
|-------|---------------|----------|---------|------------|
| Psycho-S | $\mathbf{14 \times 14}$: $[256\ (\mathbb{I}), 256, 384, 512, 512]$ | 512 | 65 / 9 | 25.35 |
| Psycho-B | $\mathbf{28 \times 28}$: $[256\ (\mathbb{I}), 256, 256, 384]$ 
 $\mathbf{14 \times 14}$: $[384, 384, 512, 512, 512]$ | 512 | 93 / 13 | 42.01 |
| Psycho-L | $\mathbf{28 \times 28}$: $[256\ (\mathbb{I}), 512, 512, 512]$ 
 $\mathbf{14 \times 14}$: $[512, 512, 512, 512, 512]$ | 512 | 93 / 13 | 61.28 |
| Psycho-H | $\mathbf{28 \times 28}$: $[256\ (\mathbb{I}), 512, 512, 512]$ 
 $\mathbf{14 \times 14}$: $[512, 512, 512, 640, 1024]$ | 1024 | 93 / 13 | 88.61 |
| Psycho-DW | $\mathbf{28 \times 28}$: $[256\ (\mathbb{I}), 256, 256, 512]$ 
 $\mathbf{14 \times 14}$: $[512, 1024, 1024, 1024, 1024]$ | 2048 | 109 / 13 | 49.512 |

Table A.3: Detailed architecture of Psycho-S.

| **Psycho-S - based on ResNet50** | |
|---|---|
| Parameters (M) | 25.35 |
| # Layers (overall) | 65 |
| # Layers (complex) | 9 |
| **Blocks** | |
| Input layer | Conv2D($7 \times 7$, $d_{\text{in}}$=3, $d_{\text{out}}$=64, stride=2), MaxPool($3 \times 3$, stride=2) |
| Initial CNN layers | First 7 ResBlocks from ResNet50 (first two resolution stages). |
| Phasor Blocks | $\mathbf{14 \times 14}$: $(\mathbb{I})$ $[d_{\text{in}}$=128, $d_{\text{out}}$=256, stride=2$]$ 
 $\mathbf{14 \times 14}$: $(\mathbb{C})$ $[d_{\text{in}}$=256, $d_{\text{out}}$=256$]$ 
 $\mathbf{14 \times 14}$: $(\mathbb{C})$ $[d_{\text{in}}$=256, $d_{\text{out}}$=384$]$ 
 $\mathbf{14 \times 14}$: $(\mathbb{C})$ $[d_{\text{in}}$=384, $d_{\text{out}}$=512$]$ 
 $\mathbf{14 \times 14}$: $(\mathbb{C})$ $[d_{\text{in}}$=512, $d_{\text{out}}$=512$]$ |
| Spectral filters | Sub-bands ([crop, drop]): [14, 8], [8, 4], [4, 1],   d_filter = 512 |
| Output layer | Average pool, ComplexLinear($d_{\text{in}}$=1536, $d_{\text{out}}$=1000), Softmax |

Table A.4: Detailed architecture of Psycho-B.

| Psycho-B architecture - based on ResNet101 | |
|---|---|
| Parameters (M) | 42.01 |
| # Layers (overall) | 93 |
| # Layers (complex) | 13 |
| **Blocks** | |
| Input layer | Conv2D($7 \times 7$, $d_{\text{in}}$=3, $d_{\text{out}}$=64, stride=2), MaxPool($3 \times 3$, stride=2) |
| Initial CNN layers | First 7 ResBlocks from ResNet101 (first two resolution stages). |
| Phasor Blocks | $\mathbf{28 \times 28}$: ($\mathbb{I}$) [$d_{\text{in}}$=128, $d_{\text{out}}$=256]
$\mathbf{28 \times 28}$: ($\mathbb{C}$) [$d_{\text{in}}$=256, $d_{\text{out}}$=256]
$\mathbf{28 \times 28}$: ($\mathbb{C}$) [$d_{\text{in}}$=256, $d_{\text{out}}$=256]
$\mathbf{28 \times 28}$: ($\mathbb{C}$) [$d_{\text{in}}$=256, $d_{\text{out}}$=384]

$\mathbf{14 \times 14}$: ($\mathbb{C}$) [$d_{\text{in}}$=384, $d_{\text{out}}$=384, stride=2]
$\mathbf{14 \times 14}$: ($\mathbb{C}$) [$d_{\text{in}}$=384, $d_{\text{out}}$=384]
$\mathbf{14 \times 14}$: ($\mathbb{C}$) [$d_{\text{in}}$=384, $d_{\text{out}}$=512]
$\mathbf{14 \times 14}$: ($\mathbb{C}$) [$d_{\text{in}}$=512, $d_{\text{out}}$=512]
$\mathbf{14 \times 14}$: ($\mathbb{C}$) [$d_{\text{in}}$=512, $d_{\text{out}}$=512] |
| Spectral filters | Sub-bands ([crop, drop]): [14, 8], [8, 4], [4, 1],     d_filter = 512 |
| Output layer | Average pool, ComplexLinear($d_{\text{in}}$=1536, $d_{\text{out}}$=1000), Softmax |

Table A.5: Detailed architecture of Psycho-L.

| Psycho-L architecture - based on ResNet152 | |
|---|---|
| Parameters (M) | 61.28 |
| # Layers (overall) | 93 |
| # Layers (complex) | 13 |
| **Blocks** | |
| Input layer | Conv2D($7 \times 7$, $d_{\text{in}}$=3, $d_{\text{out}}$=64, stride=2), MaxPool($3 \times 3$, stride=2) |
| Initial CNN layers | First 7 ResBlocks from ResNet152. |
| Phasor Blocks | $\mathbf{28 \times 28}$: ($\mathbb{I}$) [$d_{\text{in}}$=128, $d_{\text{out}}$=256]
$\mathbf{28 \times 28}$: ($\mathbb{C}$) [$d_{\text{in}}$=256, $d_{\text{out}}$=512]
$\mathbf{28 \times 28}$: ($\mathbb{C}$) [$d_{\text{in}}$=512, $d_{\text{out}}$=512]
$\mathbf{28 \times 28}$: ($\mathbb{C}$) [$d_{\text{in}}$=512, $d_{\text{out}}$=512]

$\mathbf{14 \times 14}$: ($\mathbb{C}$) [$d_{\text{in}}$=512, $d_{\text{out}}$=512, stride=2]
$\mathbf{14 \times 14}$: ($\mathbb{C}$) [$d_{\text{in}}$=512, $d_{\text{out}}$=512]
$\mathbf{14 \times 14}$: ($\mathbb{C}$) [$d_{\text{in}}$=512, $d_{\text{out}}$=512]
$\mathbf{14 \times 14}$: ($\mathbb{C}$) [$d_{\text{in}}$=512, $d_{\text{out}}$=512]
$\mathbf{14 \times 14}$: ($\mathbb{C}$) [$d_{\text{in}}$=512, $d_{\text{out}}$=512] |
| Spectral filters | Sub-bands ([crop, drop]): [14, 8], [8, 4], [4, 1],     d_filter = 512 |
| Output layer | Average pool, ComplexLinear($d_{\text{in}}$=1536, $d_{\text{out}}$=1000), Softmax |

Table A.6: Detailed architecture of Psycho-H.

| **Psycho-H architecture - based on ResNet270** | |
| --- | --- |
| Parameters (M) | 88.61 |
| # Layers (overall) | 93 |
| # Layers (complex) | 13 |
| **Blocks** | |
| Input layer | Conv2D($7 \times 7$, $d_{in}$=3, $d_{out}$=64, stride=2), MaxPool($3 \times 3$, stride=2) |
| Initial CNN layers | First 7 ResBlocks from ResNet270. |
| Phasor Blocks | $\mathbf{28 \times 28}$: ($\mathbb{I}$) [$d_{in}$=128, $d_{out}$=256]
$\mathbf{28 \times 28}$: ($\mathbb{C}$) [$d_{in}$=256, $d_{out}$=512]
$\mathbf{28 \times 28}$: ($\mathbb{C}$) [$d_{in}$=512, $d_{out}$=512]
$\mathbf{28 \times 28}$: ($\mathbb{C}$) [$d_{in}$=512, $d_{out}$=512]

$\mathbf{14 \times 14}$: ($\mathbb{C}$) [$d_{in}$=512, $d_{out}$=512, stride=2]
$\mathbf{14 \times 14}$: ($\mathbb{C}$) [$d_{in}$=512, $d_{out}$=512]
$\mathbf{14 \times 14}$: ($\mathbb{C}$) [$d_{in}$=512, $d_{out}$=512]
$\mathbf{14 \times 14}$: ($\mathbb{C}$) [$d_{in}$=512, $d_{out}$=640]
$\mathbf{14 \times 14}$: ($\mathbb{C}$) [$d_{in}$=640, $d_{out}$=1024] |
| Spectral filters | Sub-bands ([crop, drop]): [14, 8], [8, 4], [4, 1],      d_filter = 1024 |
| Output layer | Average pool, ComplexLinear($d_{in}$=3072, $d_{out}$=1000), Softmax |

Table A.7: Detailed architecture of PsychoDW.

| **PsychoDW architecture - based on ConvNeXt-S** | |
| --- | --- |
| Parameters (M) | 49.512 |
| # Layers (overall) | 109 |
| # Layers (complex) | 13 |
| **Blocks** | |
| Input layer | Conv2D($7 \times 7$, $d_{in}$=3, $d_{out}$=64, stride=2), MaxPool($3 \times 3$, stride=2) |
| Initial CNN layers | First 7 ResBlocks from ResNet50. |
| Phasor Blocks | $\mathbf{28 \times 28}$: ($\mathbb{I}$) [$d_{in}$=128, $d_{out}$=256]
$\mathbf{28 \times 28}$: ($\mathbb{C}$) [$d_{in}$=256, $d_{out}$=256]
$\mathbf{28 \times 28}$: ($\mathbb{C}$) [$d_{in}$=256, $d_{out}$=256]
$\mathbf{28 \times 28}$: ($\mathbb{C}$) [$d_{in}$=256, $d_{out}$=512]

$\mathbf{14 \times 14}$: ($\mathbb{C}$) [$d_{in}$=512, $d_{out}$=512, stride=2]
$\mathbf{14 \times 14}$: ($\mathbb{C}$) [$d_{in}$=512, $d_{out}$=1024]
$\mathbf{14 \times 14}$: ($\mathbb{C}$) [$d_{in}$=1024, $d_{out}$=1024]
$\mathbf{14 \times 14}$: ($\mathbb{C}$) [$d_{in}$=1024, $d_{out}$=1024]
$\mathbf{14 \times 14}$: ($\mathbb{C}$) [$d_{in}$=1024, $d_{out}$=1024] |
| Spectral filters | Sub-bands ([crop, drop]): [14, 8], [8, 4], [4, 1],      d_filter = 1024 |
| Output layer | Average pool, ComplexLinear($d_{in}$=3072, $d_{out}$=1000), Softmax |

## B.3 PHASOR BLOCK ARCHITECTURE

Figure A.2 provides detailed architectural diagrams of Phasor Blocks, with key design choices discussed below.

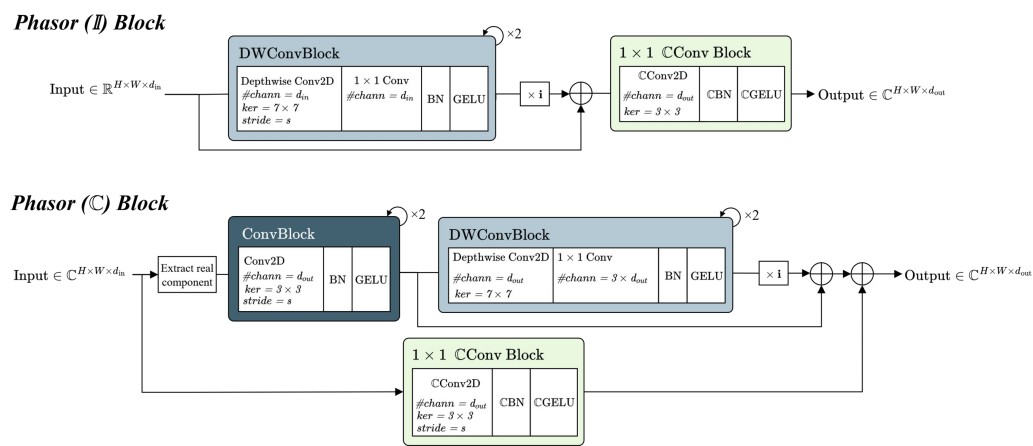

Figure A.2: Further architecture details for the Phasor Blocks presented in Figure 4. For ConvNeXt-based PsychoNet, we replace the two ConvBlocks at the start of Phasor ($\mathbb{C}$) blocks with two DW-ConvBlocks with the same number of channels. $\mathbb{C}$Conv/BN/GELU denote complex-valued convolution, batch norm and GELU operations - see Appendix A.3. The following PsychoNet architecture tables specify the values of $d_{in}$, $d_{out}$ and stride ($s$) for all of their Phasor Blocks.

Phasor ($\mathbb{I}$) blocks generate an initial set of imaginary components using depthwise convolution ('DWConv') blocks, comprising pairs of depthwise and $1 \times 1$ convolution layers. This configuration decouples spatial and channel mixing, which is intended to encourage cross-channel interactions without interfering with spatial relationships. In natural complex signals, the real and imaginary components carry complementary information for the same spatial location (Gonzalez & Woods, 2014; Lee et al., 2022), so it is likely important that our generated imaginary features do not significantly introduce new spatial information. A $1 \times 1$ complex convolution block then mixes the real and imaginary features. Subsequently, Phasor ($\mathbb{C}$) blocks further refine the complex representations. The top branch generates new real and imaginary features, while the bottom channel-mixes the original features and combines them with the new ones. For ConvNeXt-based PsychoNet, we replace Phasor ($\mathbb{C}$) 's regular convolution ('Conv') blocks with further DWConv blocks with $7 \times 7$ kernel size, matching ConvNeXt's main computational block.

# C CLASSIFICATION EXPERIMENTS

In this section we present detailed results, dataset details and training recipes for all classification experiments conducted.

## C.1 IMAGENET-1K

We use the standard large ImageNet-1K subset from (Deng et al., 2009) containing ~1.2 million training and ~50000 images for validation/testing. Table A.8 presents the training recipe used for ImageNet experiments.

Table A.8: ImageNet training recipe

| Setting | Value |
| --- | --- |
| Image size | $224 \times 224$ |
| Epochs | 90 |
| Batch size (overall, not per GPU) | 1024 |
| Loss | Cross entropy |
| Optimizer | AdamW ($\beta_1 = 0.9, \beta_2 = 0.999$) |
| Scheduler | cosine |
| Initial learning rate (LR) | $5 \cdot 10^{-4}$ |
| Warmup | warmup LR $= 10^{-6}$ , 5 epochs |
| Learning rate decay | min. LR $= 10^{-5}$, 12 epochs |
| Augmentation | resize, crop, interpolate, horizontal flip, RandAugment, MixUp, CutMix, label smoothing |
| GPU | $2\times$ NVIDIA H100: Psycho-B, ResNet101, all 'Big' sized ablation models
$2\times$ AMD MI300X: Psycho-S, ResNet50
$4\times$ AMD MI300X: All other models |

Table A.8 presents all ImageNet-1K experiment results. PsychoNet moderately improves top-1 accuracy for all ResNet baselines ($\uparrow$ 0.82%, 0.41%, 0.26% and 0.44% vs. ResNet50 to 270), and incurs a small decrease for ConvNeXt-S ($\downarrow$ 0.19). Figure A.3 compares SVC filters learnt by different ResNet-based PsychoNet sizes, showing that with larger model size, the filters become increasingly structured and sparser, with clearer frequency selectivity and reduced noise. Figure A.4 compares SVC filters learnt by Psycho-B on ImageNet-1K to the smaller resolution/size datasets in Appendix C.2. It is evident that increasing image resolution and dataset size both yield much sparser filters. These results suggest that the sparse patterns correspond to a data-driven representation naturally emergent from visual information.

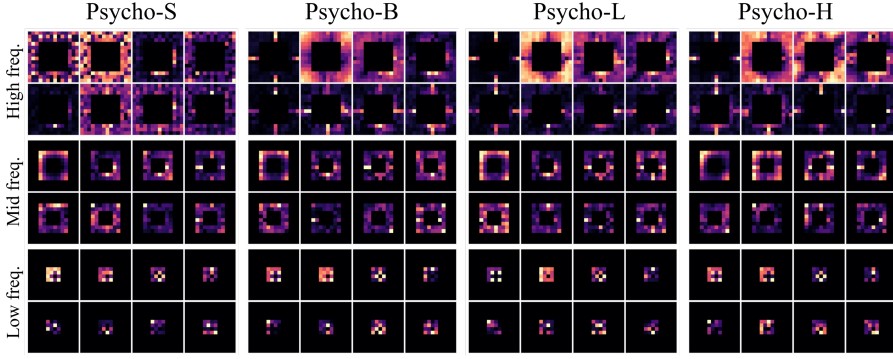

Figure A.3: Top principal components of SVC filters learnt by different sized ResNet-based PsychoNet models on ImageNet-1K. 'High/mid/low freq.' refer to the [14, 8], [8, 4] and [4, 1] frequency sub-bands created by Spectral Branches.

Table A.9: ImageNet-1K classification results. Each pair of rows (separated by horizontal lines) compares a baseline CNN and the PsychoNet based on it. FLOPs were measured using a single $224 \times 224$ input.

| Model | Top-1 Acc. (%) | Top-5 Acc. (%) | Layers | Params (M) | FLOPs (G) | GPU |
|---|---|---|---|---|---|---|
| ResNet50 | 76.044 | 92.992 | 54 | 25.56 | 8.18 | 2× MI300X |
| Psycho-S | 76.864 | 93.386 | 65 | 25.35 | 12.31 | 2× MI300X |
| ResNet101 | 78.428 | 94.220 | 105 | 44.55 | 15.60 | 2× H100 |
| Psycho-B | 78.846 | 94.600 | 93 | 42.01 | 30.13 | 2× H100 |
| ResNet152 | 79.586 | 94.684 | 156 | 60.19 | 23.03 | 4× MI300X |
| Psycho-L | 79.848 | 95.056 | 93 | 61.28 | 54.47 | 4× MI300X |
| ResNet270 | 80.012 | 95.088 | 276 | 89.60 | 40.50 | 4× MI300X |
| Psycho-H | 80.454 | 95.290 | 93 | 88.61 | 64.12 | 4× MI300X |
| ConvNeXt-S | 80.780 | 95.488 | 113 | 50.22 | 17.36 | 2× MI300X |
| Psycho-DW | 80.590 | 95.384 | 106 | 49.51 | 27.42 | 2× MI300X |

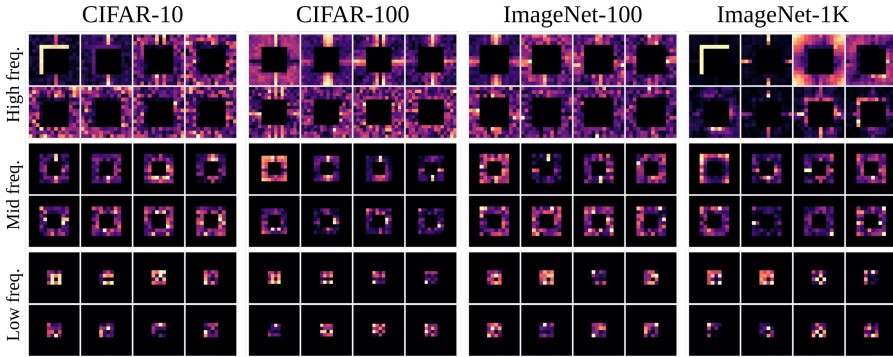

Figure A.4: Top principal components of SVC filters learnt by Psycho-B on different resolution and size datasets. 'High/mid/low freq.' refer to the [14, 8], [8, 4] and [4, 1] frequency sub-bands created by Spectral Branches.

### C.1.1 PSYCHODW REPRESENTATION ANALYSIS

Figures A.5 through A.7 present qualitative visualisations and analysis for PsychoDW identical to those applied to Psycho-B we presented in Section 4. Overall, these show similar results:

- Figure A.5 **(a)** shows that PsychoDW's SVC filters also learn sparse selections of frequencies across each sub-band.
- Figure A.5 **(b)** shows that similar to the Psycho-B vs. ResNet-101 comparison in Figure 5 **(b)**, salience maps of PsychoDW's low-mid level Phasor Blocks clearly emphasis specific object parts, while those of ConvNeXt-S are much more general and diffuse. Further examples of the former are shown in Figure A.6.
- Figure A.7 shows that similar to for Psycho-B in Figure 8, PsychoDW's SVC appears to distribute object parts by scale between the three sub-bands, and individual filters within each sub-band target distinct selections of object parts.

Overall, these results are highly consistent with those for Psycho-B, showing that SVC abstractions and object-part-centric Phasor Block representations also translate to ConvNeXt-S.

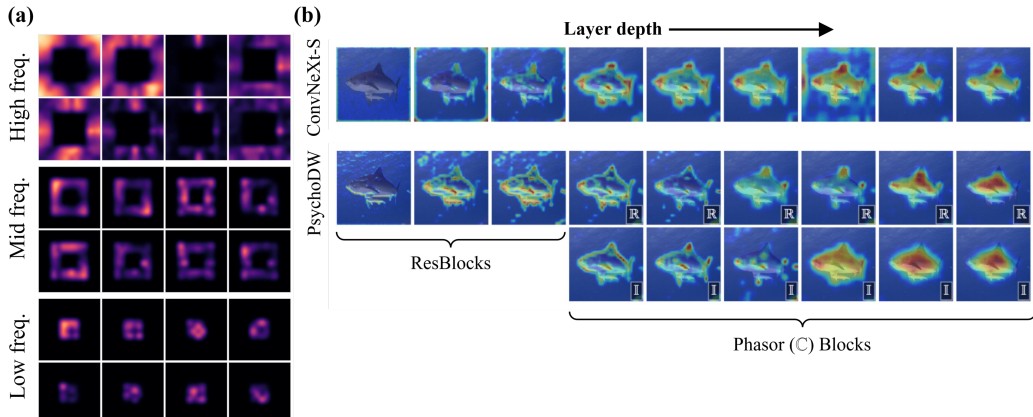

Figure A.5: **(a)** Top spatial principal components of SVC filters learnt by PsychoDW trained on ImageNet-1K. Bilinear smoothing has been applied. **(b)** Comparison between activation maps (via KPCA-CAM) of PsychoDW and ConvNeXt-S for a range of layer depths. Real and imaginary components are denoted by $\mathbb{R}$ and $\mathbb{I}$.

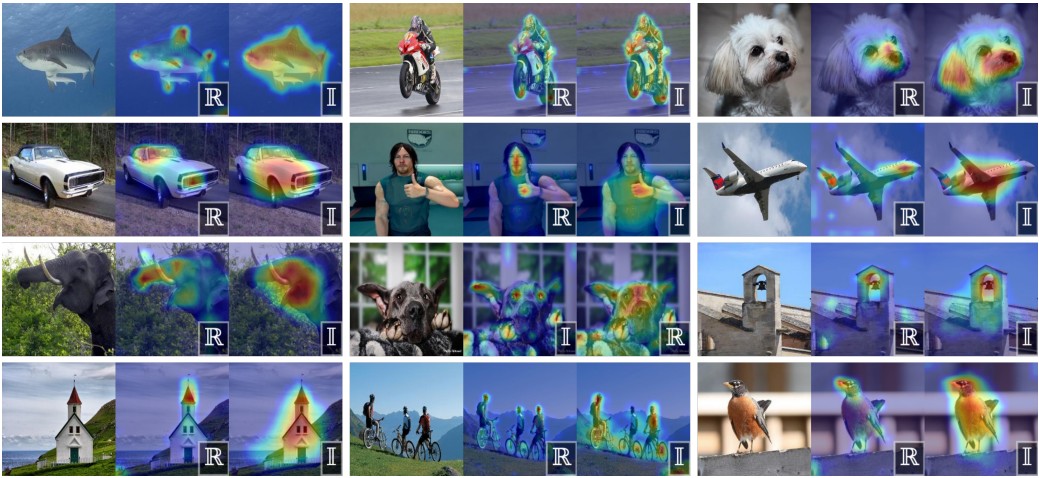

Figure A.6: Assorted activation maps (via KPCA-CAM) for mid-level Phasor Blocks of PsychoDW. Real and imaginary components are denoted by $\mathbb{R}$ and $\mathbb{I}$.

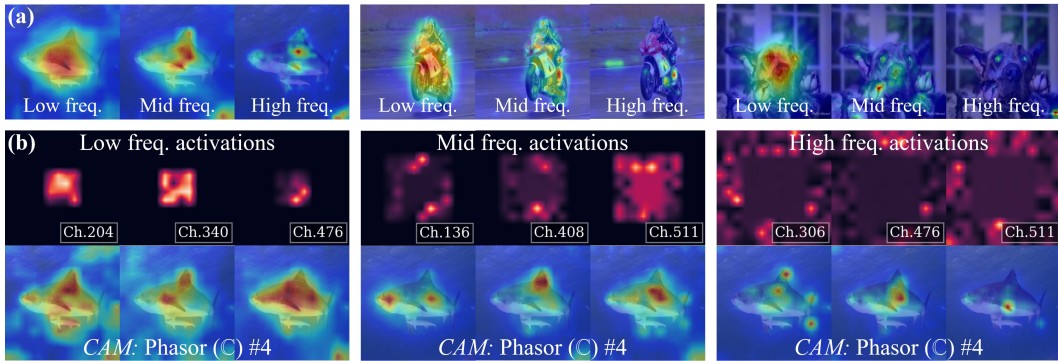

Figure A.7: PsychoDW Phasor Block salience maps (via HiResCAM) conditioned on gradients **(a)** from individual Spectral Branch sub-bands and **(b)** from individual frequency domain feature channels.

## C.2 SMALLER CLASSIFICATION DATASETS

Table A.10 presents experiment results for the CIFAR-10, CIFAR-100 and ImageNet-100 classification experiments.

| Module | Parameters (M) | # Layers | CIFAR-10 | CIFAR-100 | ImageNet-100 |
|---|---|---|---|---|---|
| ResNet50 | 25.56 | 54 | 94.14 | 78.10 | 80.90 |
| Psycho-S | 25.35 | 65 | 95.08 | 78.97 | 82.50 |
| ResNet101 | 44.55 | 105 | 93.64 | 79.13 | 81.90 |
| Psycho-B | 42.01 | 95 | 94.99 | 79.49 | 83.60 |
| ResNet152 | 60.10 | 156 | 93.17 | 77.51 | 83.60 |
| Psycho-L | 61.28 | 93 | 94.95 | 79.64 | 84.82 |
| ResNet270 | 89.60 | 276 | 76.51 | 50.87 | 83.80 |
| Psycho-H | 88.61 | 93 | 94.68 | 79.89 | 85.00 |
| ConvNeXt-S | 50.22 | 113 | 94.09 | 76.96 | 86.98 |
| PsychoDW | 49.51 | 106 | 95.46 | 79.67 | 86.76 |

Table A.10: Classification results (% top-1 accuracies) for CIFAR-10, CIFAR-100 and ImageNet-100. Each pair of rows (separated by horizontal lines) compares a baseline CNN and the PsychoNet based on it.

**CIFAR-10** is a small scale dataset comprising 50000 natural images for training and 10000 images for testing across 10 classes, at a resolution of $32 \times 32$ (Krizhevsky, 2009). For compatibility with this lower resolution (the ImageNet models have $224 \times 224$ input resolution), we reduce initial downsampling steps from our models. For ResNet and ResNet-based PsychoNet models, we removed the first maxpooling layer and set stride=1 for the first two ResBlocks that originally had stride=2. For ConvNeXt-S and PsychoDW, we replace the initial $4 \times 4$ patch embedding layer with a standard $3 \times 3$ Conv2D layer, and set stride=1 for the second downsampling layer. Table A.11 presents the training recipe for the CIFAR-10 experiments. Overall, all of our PsychoNet models outperformed their respective CNN baselines.

**CIFAR-100** contains the same images and train-test split as CIFAR-10, but with labels reorganised into 100 classes instead of 10. We use the same model configurations and training recipe as CIFAR-10, but increase the number of epochs to 90 since the greater number of classes results in a harder classification problem. Table A.11 presents the training recipe for the CIFAR-10 experiments. Overall, all of our PsychoNet models outperformed their respective CNN baselines.

**ImageNet-100** is a subset of the ImageNet dataset (Deng et al., 2009) that contains examples for 100 classes. It contains 130100 images for training and 5100 images for testing, at the original

Table A.11: CIFAR-10 training recipe

| Setting | Value |
| --- | --- |
| Image size | $32 \times 32$ |
| Epochs | 35 |
| Batch size | 64 |
| Loss | Cross entropy |
| Optimizer | AdamW ($\beta_1 = 0.9, \beta_2 = 0.999$) |
| Scheduler | OneCycle |
| Learning rate (LR) | $10^{-3}$ |
| Augmentation | crop, horizontal flip |
| GPU | $1\times$ NVIDIA A100: Psycho-S/B, ResNet50/101
$1\times$ NVIDIA H100: All other models |

resolution of $224 \times 224$. The model architectures remain the same as the ImageNet experiments, but with the output linear layer modified to predict 100 logits. We use the same training recipe as ImageNet-1K (Table A.8), but reduce the batch size to 128. Psycho-S/B and ResNet50/101 were trained on $1\times$ NVIDIA A100, while all over models used $1\times$ AMD MI300X. Overall, the ResNet-based PsychoNet models outperformed their respective baselines, but PsychoDW fell slightly short of ConvNeXt-S.

### C.3 CLUSTERING VISUALISATION

Figure A.8 presents an initial visualisation of clustering characteristics of Phasor Block activations and SVC for Psycho-B. These show 2D PCA projections of features computed on samples from 10 randomly-selected classes from ImageNet-1K. Observable clustering emerges across both the real and imaginary/magnitude-phase feature components, and becomes increasingly pronounced at deeper layers.

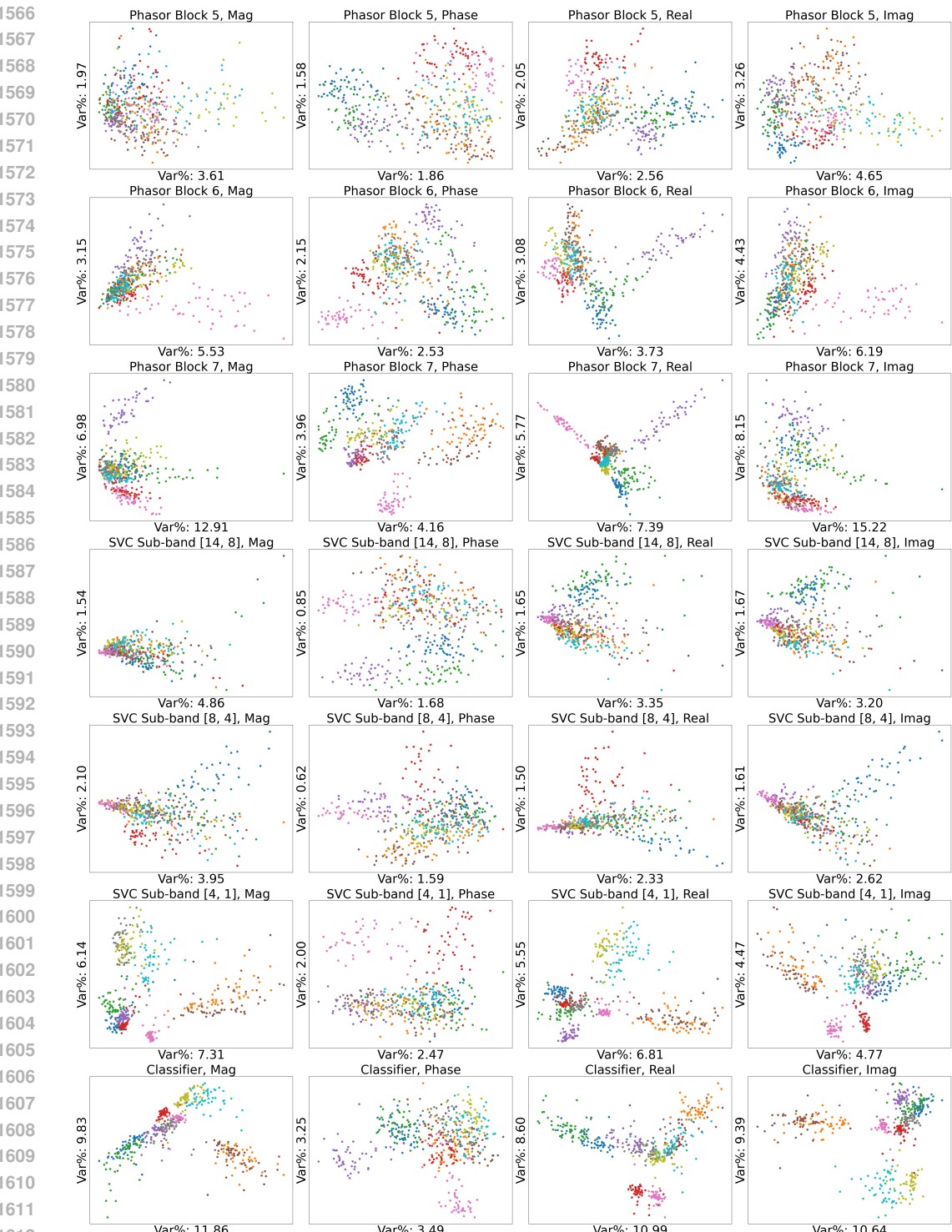

Figure A.8: Psycho-B clustering visualisation.

# D ABLATION STUDIES

## D.1 ARCHITECTURAL ABLATIONS

We design four ablation model configurations to assess the impact of Phasor Blocks and SVC's multiple Spectral Branches on classification performance and visual code quality.

The postfix SP (Single Phasor) indicates that we remove all Phasor Blocks ($\mathbb{C}$)s and make up for the resultant layer and parameter deficit by adding additional ResBlocks. SP_MB (Big/Large) were created by applying this modification to Psycho-B/L respectively, and in-depth architectural details of them are presented in Tables A.13 and A.14. The Single Branch (SB) models replace Spectral Branches with a single Hadamard Block with full band filters and no prior DropCrop operations. MP_SB (Big/Large) and SP_SB (Big/Large) were created by applying this modification to Psycho-B/L and SP_MB (Big/Large) respectively. Table A.12 and Figure A.9 present quantitative and qualitative results from this study.

Spectral Branches appreciably improve classification accuracy (0.45-0.58%), except for between Big size SP_MB and SP_SB. In Figure A.9, we visualize the first two spectral bands of the MB models, as well as the corresponding bands isolated from the full-band filters of the SB models. The former are sparse and highlight distinct frequencies, while the latter exhibit a similar structure but with significant noise. This suggests that the explicit spectral decomposition of Spectral Branches is important for generating clear visual codes. Multiple Phasor Blocks slightly improve classification accuracy (0.24-0.252%) for MB models and have little effect on SB ones. However, Figure A.9 shows that they drastically reduce noise and improve clarity of the SB filters, and moderately so for the MB ones.

Finally, we also try removing the Phasor ($\mathbb{I}$) block from the Big size SP_MB model, yielding a model without any Phasor Blocks ('no_phasor'). This further reduces accuracy slightly, and results in the filters exhibiting conjugate symmetry as shown in Figure 6. We further visualise salience maps for the no_phasor ablation in Figure A.10. Compared to ResNet, the no_phasor model places greater emphasis on edge information across intermediate layers, likely reflecting the downstream SVC making explicit use of high-frequency features. In later layers, salience increasingly concentrates on object parts, including the shark's head and fins, consistent with our broader evidence that SVC encourages object-part-based abstraction (Section 4). However, when Phasor Blocks are included—as in the full Psycho-B model—object parts are isolated far more clearly in the salience maps. This suggests that while SVC promotes part-focused semantic representations, the complex representations introduced by Phasor Blocks are important for expressing these abstractions cleanly and with high specificity.

Table A.12: **Ablation study results.** We compare all combinations of MB/SB and MP/SP model configurations, for Big and Large model sizes, using ImageNet top-1 accuracy.

| Model | **M**ultiple Phasor **B**locks | **M**ultiple (Spectral) **B**ranches | Top-1 Acc. (%) (Psycho-B base) | Top-1 Acc. (%) (Psycho-L base) |
|---|---|---|---|---|
| MP_MB (Psycho-B/L) | ✓ | ✓ | 78.846 | 79.848 |
| MP_SB | ✓ | ✗ | 78.394 | 79.268 |
| SP_MB | ✗ | ✓ | 78.600 | 79.596 |
| SP_SB | ✗ | ✗ | 78.548 | 79.124 |
| no_phasor (SP_SB w/o Phasor ($\mathbb{I}$) ) | ✗ | ✗ | 78.44 | |

Note that as per He et al. (2016), ResBlocks each comprises $1 \times 1$, $3 \times 3$ and $1 \times 1$ kernel size Conv2D layers. In the below architecture tables, we denote their respective output channel sizes with $d_{\text{in}}$, $d_{\text{bot}}$ and $d_{\text{out}}$ respectively ('bot' is short for bottleneck, as these layers follow a channel bottleneck configuration). We also write 'stride=2' if a ResBlock performs $2\times$ spatial downsampling, since it is achieved by setting stride=2 in the $3 \times 3$ Conv2D layer.

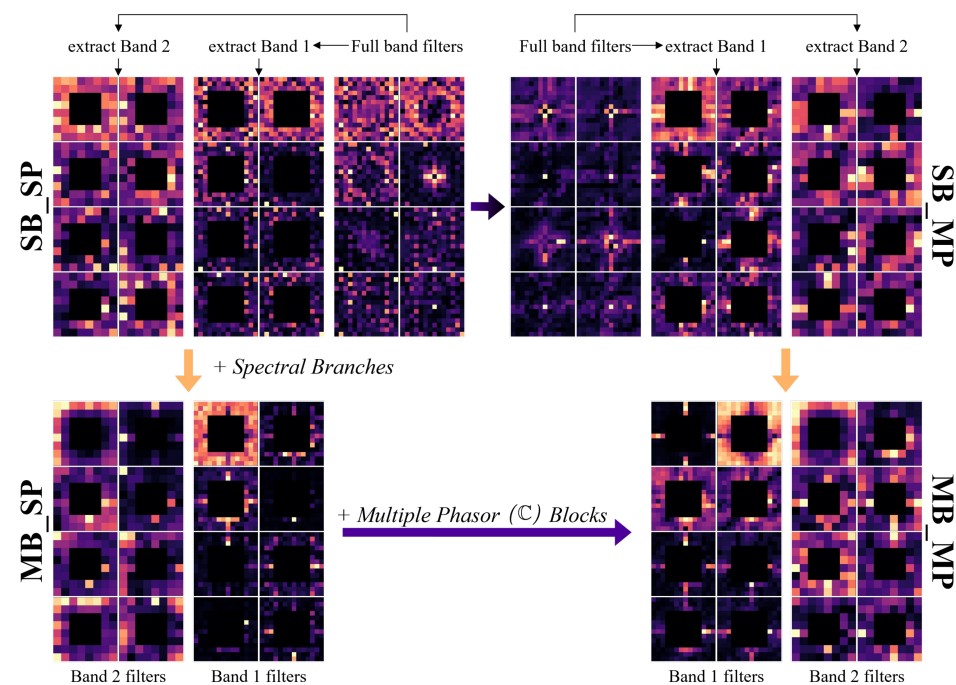

Figure A.9: Most significant channel-wise principal components of learnt spectral filters from Large size ablation models. We show the first two sub-bands of filters for MB models, and the full band filter for SB models.

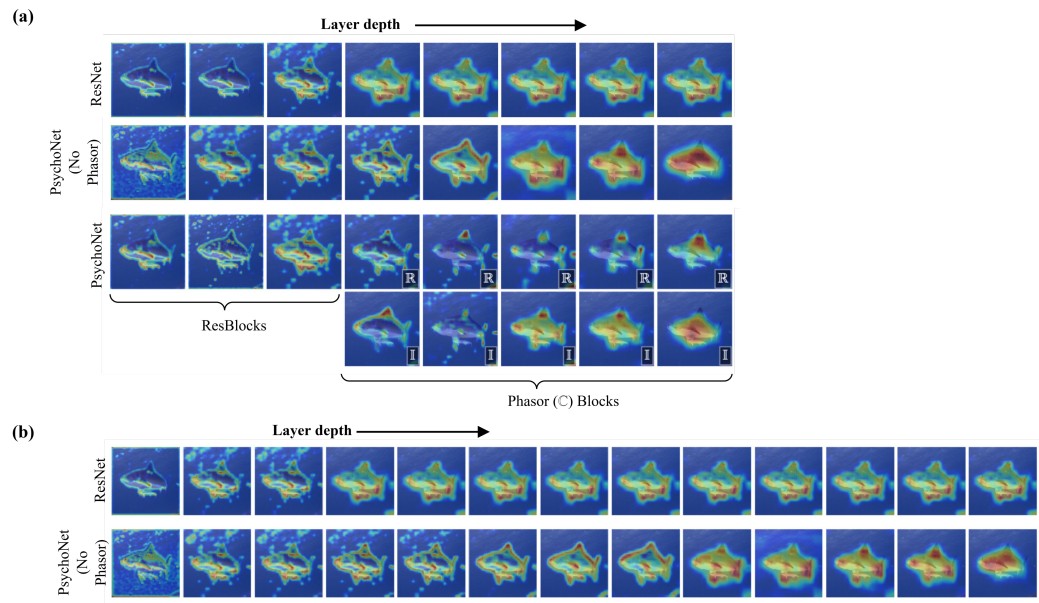

Figure A.10: **(a)** An extended version of Figure 5 (b) which additionally adds KPCACAM salience maps for the *no_phasor* ablation model. **(a)** Additional salience maps comparison between ResNet-101 and *no_phasor*.

### D.1.1 FLOP EFFICIENT MODELS

As we identified that PsychoNet used considerably more FLOPs than ResNet and ConvNeXt baselines (see Table A.12), we conducted the following ablation experiment to show a fairly straightfor-

Table A.13: Detailed architecture of the SP_MB (Big) ablation model.

| SP_MB (Big) architecture | |
|---|---|
| Parameters (M) | 42.263 |
| # Layers (overall) | 91 |
| # Layers (complex) | 2 |
| **Blocks** | |
| Input layer | Conv2D($7 \times 7$, $d_\text{in}$=3, $d_\text{out}$=64, stride=2), MaxPool($3 \times 3$, stride=2) |
| ResBlocks | $\mathbf{56 \times 56}$: $[d_\text{in}$=64, $d_\text{bot}$=256, $d_\text{out}$=256$]$ 
 $\mathbf{56 \times 56}$: $[d_\text{in}$=256, $d_\text{bot}$=64, $d_\text{out}$=256$] \times 2$ 
 $\mathbf{28 \times 28}$: $[d_\text{in}$=256, $d_\text{bot}$=128, $d_\text{out}$=512, stride=2$]$ 
 $\mathbf{28 \times 28}$: $[d_\text{in}$=512, $d_\text{bot}$=128, $d_\text{out}$=512$] \times 7$ 
 $\mathbf{28 \times 28}$: $[d_\text{in}$=512, $d_\text{bot}$=256, $d_\text{out}$=1024$]$ 
 $\mathbf{28 \times 28}$: $[d_\text{in}$=1024, $d_\text{bot}$=256, $d_\text{out}$=1024$] \times 4$ 
 $\mathbf{14 \times 14}$: $[d_\text{in}$=1024, $d_\text{bot}$=256, $d_\text{out}$=1024, stride=2$]$ 
 $\mathbf{14 \times 14}$: $[d_\text{in}$=1024, $d_\text{bot}$=384, $d_\text{out}$=1536$]$ 
 $\mathbf{14 \times 14}$: $[d_\text{in}$=1536, $d_\text{bot}$=384, $d_\text{out}$=1536$] \times 6$ 
 $\mathbf{14 \times 14}$: $[d_\text{in}$=1536, $d_\text{bot}$=128, $d_\text{out}$=512$]$ |
| Phasor Blocks | $\mathbf{14 \times 14}$: $(\mathbb{I})$ $[d_\text{in}$=512, $d_\text{out}$=512, stride=1$]$ |
| SVC filters | Sub-bands ([crop, drop]): [14, 8], [8, 4], [4, 1],     $d_\text{filter}$ 512 |
| Output layer | Average pool, ComplexLinear($d_\text{in}$=1536, $d_\text{out}$=1000), Softmax |

Table A.14: Detailed architecture of the SP_MB (Large) ablation model.

| SP_MB (Large) architecture | |
|---|---|
| Parameters (M) | 60.42 |
| # Layers (overall) | 90 |
| # Layers (complex) | 2 |
| **Blocks** | |
| Input layer | Conv2D($7 \times 7$, $d_\text{in}$=3, $d_\text{out}$=64, stride=2), MaxPool($3 \times 3$, stride=2) |
| ResBlocks | $\mathbf{56 \times 56}$: $[d_\text{in}$=64, $d_\text{bot}$=256, $d_\text{out}$=256$]$ 
 $\mathbf{56 \times 56}$: $[d_\text{in}$=256, $d_\text{bot}$=64, $d_\text{out}$=256$] \times 2$ 
 $\mathbf{28 \times 28}$: $[d_\text{in}$=256, $d_\text{bot}$=128, $d_\text{out}$=512, stride=2$]$ 
 $\mathbf{28 \times 28}$: $[d_\text{in}$=512, $d_\text{bot}$=128, $d_\text{out}$=512$] \times 5$ 
 $\mathbf{28 \times 28}$: $[d_\text{in}$=512, $d_\text{bot}$=256, $d_\text{out}$=1024$]$ 
 $\mathbf{28 \times 28}$: $[d_\text{in}$=1024, $d_\text{bot}$=256, $d_\text{out}$=1024$] \times 6$ 
 $\mathbf{14 \times 14}$: $[d_\text{in}$=1024, $d_\text{bot}$=512, $d_\text{out}$=2048, stride=2$]$ 
 $\mathbf{14 \times 14}$: $[d_\text{in}$=2048, $d_\text{bot}$=512, $d_\text{out}$=2048$] \times 7$ 
 $\mathbf{14 \times 14}$: $[d_\text{in}$=2048, $d_\text{bot}$=128, $d_\text{out}$=512$]$ |
| Phasor Blocks | $\mathbf{14 \times 14}$: $(\mathbb{I})$ $[d_\text{in}$=512, $d_\text{out}$=512, stride=1$]$ |
| SVC filters | Sub-bands ([crop, drop]): [14, 8], [8, 4], [4, 1],     $d_\text{filter}$ 512 |
| Output layer | Average pool, ComplexLinear($d_\text{in}$=1536, $d_\text{out}$=1000), Softmax |

ward architectural modification can be applied to improve FLOP efficiency of PsychoNet. Models Psycho-S-7×7 and Psycho-B-7×7 are variations of Psycho-S and Psycho-B which decrease the spatial feature size of some Phasor Block layers, obtained via:

- We add an additional downsampling stage by setting stride $= 2$ in one of the $14 \times 14$ spatial feature size Phasor Blocks, so that the subsequent layers in the model now have a spatial feature size of $7 \times 7$.

- To preserve $14 \times 14$ SVC filters, input features to sub-bands covering features from $7 \times 7$ to $14 \times 14$ sub-band are taken from the output of the $14 \times 14$ Phasor Block immediately preceding the new downsampling stage.

As shown in Table A.15, both Psycho-S-7×7 and Psycho-B-7×7 slightly underperform the original PsychoNet models, but match the much lower FLOP counts of the baseline ResNets and beat their performance. This suggests that exploiting higher-resolution features from earlier stages while reducing spatial resolution in deeper models is an effective and scalable strategy for improving efficiency, which will be investigated further in future work. Full architectural configurations for Psycho-S-7×7 and Psycho-B-7×7 are presented in Tables A.16 and A.17 respectively.

| Model | Parameters (M) | # Layers | FLOPs (G) | ImageNet-100 Top-1 Acc (%) |
|---|---|---|---|---|
| ResNet50 | 23.71 | 54 | 8.18 | 80.24 |
| Psycho-S | 22.59 | 62 | 12.30 | 82.50 |
| Psycho-S-7×7 | 24.80 | 62 | 8.17 | 82.34 |
| ResNet101 | 42.71 | 105 | 15.60 | 82.58 |
| Psycho-B | 39.24 | 90 | 30.12 | 83.60 |
| Psycho-B-7×7 | 38.92 | 90 | 12.97 | 83.50 |

Table A.15: Comparison of FLOP-efficient Psycho-7×7 models against PsychoNet and ResNet baselines on ImageNet-100 classification.

Table A.16: Detailed architecture of Psycho-S-7×7.

| **Psycho-S-7×7** | |
|---|---|
| Parameters (M) | 24.80 |
| # Layers (overall) | 62 |
| # Layers (complex) | 9 |
| GFLOPs | 8.17 |
| **Blocks** | |
| Input layer | Conv2D($7 \times 7$, $d_{\text{in}}$=3, $d_{\text{out}}$=64, stride=2), MaxPool($3 \times 3$, stride=2) |
| Initial CNN layers | ResNet50 stem and first two resolution stages: $3\times$ ResBlocks at $56 \times 56$, $4\times$ ResBlocks at $28 \times 28$ |
| Phasor Blocks | **$28 \times 28$**: ($\mathbb{I}$) [$d_{\text{in}}$=128, $d_{\text{out}}$=256, stride=2]
**$14 \times 14$**: ($\mathbb{C}$) [$d_{\text{in}}$=256, $d_{\text{out}}$=256]
**$14 \times 14$**: ($\mathbb{C}$) [$d_{\text{in}}$=256, $d_{\text{out}}$=384]
    $uparrow$ *features used for* $14 \times 14$ *SVC input*
**$7 \times 7$**: ($\mathbb{C}$) [$d_{\text{in}}$=384, $d_{\text{out}}$=512, stride=2]
**$7 \times 7$**: ($\mathbb{C}$) [$d_{\text{in}}$=512, $d_{\text{out}}$=512] |
| SVC filters | $14 \times 14$ sub-bands ([crop, drop]): [14, 8],    $d_{\text{filter}} = 384$
$7 \times 7$ sub-bands: [7, 4], [4, 1],    $d_{\text{filter}} = 512$ |
| Output layer | Average pool, ComplexLinear($d_{\text{in}}$=1408, $d_{\text{out}}$=1000), Softmax |

Table A.17: Detailed architecture of Psycho-B-7×7.

| Psycho-B-7×7 | |
| --- | --- |
| Parameters (M) | 38.92 |
| # Layers (overall) | 90 |
| # Layers (complex) | 13 |
| GFLOPs | 12.97 |
| **Blocks** | |
| Input layer | Conv2D($7 \times 7$, $d_{\text{in}}$=3, $d_{\text{out}}$=64, stride=2), MaxPool($3 \times 3$, stride=2) |
| Initial CNN layers | ResNet101 stem and first two resolution stages: $3\times$ ResBlocks at $56 \times 56$, $4\times$ ResBlocks at $28 \times 28$ |
| Phasor Blocks | $\mathbf{28 \times 28}$: ($\mathbb{I}$) [$d_{\text{in}}$=128, $d_{\text{out}}$=256, stride=2]
$\mathbf{14 \times 14}$: ($\mathbb{C}$) [$d_{\text{in}}$=256, $d_{\text{out}}$=256]
$\mathbf{14 \times 14}$: ($\mathbb{C}$) [$d_{\text{in}}$=256, $d_{\text{out}}$=256]
$\mathbf{14 \times 14}$: ($\mathbb{C}$) [$d_{\text{in}}$=256, $d_{\text{out}}$=384]
$\mathbf{14 \times 14}$: ($\mathbb{C}$) [$d_{\text{in}}$=384, $d_{\text{out}}$=384]
$\mathbf{14 \times 14}$: ($\mathbb{C}$) [$d_{\text{in}}$=384, $d_{\text{out}}$=384]
$\uparrow$ *features used for* $14 \times 14$ *SVC input*,
$\mathbf{7 \times 7}$: ($\mathbb{C}$) [$d_{\text{in}}$=384, $d_{\text{out}}$=512]
$\mathbf{7 \times 7}$: ($\mathbb{C}$) [$d_{\text{in}}$=512, $d_{\text{out}}$=512]
$\mathbf{7 \times 7}$: ($\mathbb{C}$) [$d_{\text{in}}$=512, $d_{\text{out}}$=512] |
| SVC filters | $14 \times 14$ sub-bands ([crop, drop]): [14, 8], $d_{\text{filter}} = 384$
$7 \times 7$ sub-bands: [7, 4], [4, 1], $d_{\text{filter}} = 512$ |
| Output layer | Average pool, ComplexLinear($d_{\text{in}}$=1408, $d_{\text{out}}$=1000), Softmax |

