# OpenReview forum: "Learning frequency domain codes for semantic vision"
_ICLR.cc/2026/Conference — Submitted to ICLR 2026_

### Official Review · Reviewer_oc5W · 2025-10-27

**Soundness:** 2
**Presentation:** 3
**Contribution:** 2
**Rating:** 2
**Confidence:** 3

**Summary:**

This paper introduces Semantic Visual Coding (SVC), a learned frequency domain representation for encoding high-level visual features in convolutional neural networks. The authors develop PsychoNet, an architectural framework that adapts ResNet and ConvNeXt models to operate in both spatial and frequency domains, inspired by psychovisual processing concepts from Saadane et al. (1998). PsychoNet employs spatial layers for low-level feature extraction and frequency domain processing via SVC for high-level abstraction and reasoning. The framework is evaluated on CIFAR-10, CIFAR-100, ImageNet-100, and ImageNet-1K classification tasks, where it achieves comparable or slightly improved performance relative to baseline ResNet models, though it underperforms slightly against ConvNeXt-S on ImageNet benchmarks. Analysis of learned representations reveals that SVC converges to sparse, data-driven frequency patterns, while spatial layers extract interpretable object parts.

**Strengths:**

**Novel architectural contributions:** The paper introduces a dual-domain processing framework that separates low-level image features extraction from high-level frequency domain abstraction through what they name the PsychoNet architecture.

**Comprehensive background and contextualization:** The introduction provides thorough coverage of related work, effectively positioning the contributions within the existing literature.

**Detailed architectural exposition:** Substantial space is dedicated to explaining the components of PsychoNet.

**Reproducibility commitment:** The authors commit to releasing code including training scripts, model weights, and instructions.

**Weaknesses:**

**Unclear practical motivation and overstated contributions:** The domain or application that would benefit from this work remains unclear throughout the paper. The introduction overstates contributions by claiming:
- PsychoNet "maintains or improves the performance of common and state-of-the-art CNNs" (line 86), when performance improvements are limited to ResNet baselines and the model performs on par with ConvNeXt-S.
- "SVC performs abstraction and reasoning in the frequency domain" (line 88) contradicting the limitations section acknowledgment that "it is not clear how these representations are used for reasoning, which remains an important direction for future work" (line 430).

**Weak foundational work:** Saadane et al. (1998), the foundational psychovisual work upon which this paper builds, has only 8 citations, with 5 from the same authors. This raises questions about the influence and validation of the underlying psychovisual framework.

**Limited comparison with related work:** Lin et al. (2023), identified as "the closest work to ours" (line 138), applies Deep Frequency Filtering (DFF) to achieve state-of-the-art results on multiple domain generalization tasks including closed-set classification and open-set retrieval. In contrast, this paper only demonstrates improvements over the obsolete ResNet architecture on a single task (classification), making its contributions appear more limited in scope and impact.

**Imbalanced architectural focus:** Phasor Blocks are introduced to replace a subset of RestNet's spatial layers to break the symmetry of the Fourier Transform (FT). Specifically, Phasor Blocks augment real-valued spatial features with complementary complex-valued ones. They receive disproportionate attention in the main text. Meanwhile, DWConv Blocks used for ConvNeXt adaptation are entirely absent from the main paper discussion. The emphasis on ResNet, rather than modern networks like ConvNeXt, further limits the work's relevance.

**Incomplete resolution of stated objectives:** The limitations section explicitly states: "The key limitation of our work is that though we show SVC organises and encodes selections of object components, it is not clear how these representations are used for reasoning". Since understanding how frequency domain representations enable reasoning was presented as a primary motivation (lines 88, 92), the core contributions remain ambiguous.

**Minor:**
- Emphasis is put on the number of layers in PsychoNet models being lower than their respective baselines, yet parameter counts remain similar.
- Line 451's claim that "This pipeline mimics intermediate abstractions used by the brain to separate feature extraction from higher cognition" requires neuroscience citations to support the biological plausibility.

**Questions:**

See weaknesses section above. Also:

- ResNet270 shows worse performance than ResNet152 on the same datasets, which is unexpected. Do you have a possible explanation?
- Figure 7 lacks interpretation of why imaginary features activate on whole objects rather than object parts like real features.

---

> ### Author Response · Authors · 2025-11-25
> **Reponse to review oc5W (part 1)**
>
> We sincerely thank the reviewer for their thoughtful and constructive feedback.
>
> > Unclear practical motivation and overstated contributions: The domain or application that would benefit from this work remains unclear throughout the paper.
>
> Our work is to provide the first exploration of psychovisual processing in deep learning, an approach that models structured, abstraction-driven reasoning aligned with mechanisms of human perception described in neuroscience. Developing psychovisual principles for deep learning-based vision models could simultaneously provide interpretable data-driven functions as output alongside solid performance, which we believe will be important in majority of practical applications involving human interaction (such as medical imaging) in future. Moreover, our SVC introduces natural frequency domain representations for high-level semantic information, that will provide more transparent processing into application areas where the frequency domain is the native measurement space, such as magnetic resonance imaging (MRI). We have updated our Introduction and Future Work sections to mention this as a key area of future study.
>
> As this is the first step toward such a framework, we acknowledge that we only evaluated natural image classification, but the focus of this initial work was to verify that a psychovisual approach is feasible within deep learning and to analyse the learned representations to understand advantages they offer. Establishing broader applicability and performance across different domains will be a key direction for future work.
>
> > "SVC performs abstraction and reasoning in the frequency domain" (line 88) contradicting the limitations section acknowledgment that "it is not clear how these representations are used for reasoning, which remains an important direction for future work" (line 430).
>
> We apologise for the poor phrasing of the second sentence. Our intention was only to convey that understanding the deeper semantic meaning of abstractions produced by SVC and how they relate to broader notions of "reasoning", such as in neuroscience, is an important open question for future work. We have revised the sentence to:
>
> *“A key limitation of our work is that, although we show SVC organises and encodes selections of object components, future research is still required to determine how the deeper semantic meaning of these abstractions should be interpreted in relation to broader notions of reasoning, such as those studied in neuroscience.”*
>
> We believe that the current work already provides promising evidence that SVC enables psychovisual-like processing within CNNs and provides them with significantly more interpretable reasoning signals. Compared to the diffuse and semantically vague activation maps of ResNet, PsychoNet’s Phasor Blocks consistently focus on clear, semantically meaningful object parts for each class (Figures 5(b) and 7). Figures 6 and 8 further show that SVC learns sparse frequency-domain abstractions that (1) organise object parts by scale across sub-bands and (2) encode specific selections of parts within each filter. These learnt abstractions inform the model’s classification decisions and demonstrate that frequency domain representations provide a coherent and interpretable basis for how the model reasons.
>
> > Weak foundational work
>
> We apologise for only including one reference initially for psychovisual coding in the Background (Line 112), however there is a long history of such work. Dominique Barba's group at Nantes, France from 1991 for the following decade would publish several work in understanding the human aspect of mammalian vision, i.e. the psychovisual capability of the human brain for visual perception. They even conducted psychophysical experiments to select the optimal sub-bands that allowed image compression that was difficult for humans to distinguish. We have improved the introduction and the Background, as well as added the decade-long body of research on biologically inspired psychovisual quantizers in the 1990s to Appendix A.1 that have an additional 12 references.

---

> ### Author Response · Authors · 2025-11-25
> **Reponse to review oc5W (part 2)**
>
> > Limited comparison with related work
>
> Our description of the Lin et al. (2023)'s work as "the closest work to ours" on Line 138 was misleading and we apologize. This was only with respect to their use of frequency filters for feature modulation being similar in part to our SVC than other solely efficiency-oriented works like GFNet that are used to implement efficient global convolution/token mixers for transformer architectures not the focus of our present work. Beyond this similar aspect, Lin et al. (2023) primarily targets domain generalization performance and is not framed as representation learning or used for interpretability. Likewise, most prior frequency-domain networks are either performance-driven or used to analyse spatial model properties, so we believe that direct experimental comparisons to them would not produce meaningful insight for our study. We have revised Section 2. Frequency Domain Learning to more clearly contextualise SVC relative to prior work.
>
> > Performance improvements are limited to ResNet baselines and the model performs on par with ConvNeXt-S / Imbalanced architectural focus
>
> Our quantitative evaluation uses ResNet as a straightforward baseline for studying depth scaling, and ConvNeXt-S as a representative state-of-the-art CNN. A common characteristic of modern CNN architectures like ConvNeXt is their reduced dependence on increasing layer depth for scaling compared to older ResNets. This is due to various architectural refinements, such as ConvNeXt’s incorporation of design elements from transformers. Our results show that SVC similarly reduces the depth-dependence of ResNet, suggesting that its frequency-domain abstractions replace additional high-level processing from adding further spatial layers. While we make only modest improvements over ResNet performance, PsychoNet provides substantially clearer interpretability through structured, part-focused abstractions as shown in Figures 5 and 7. Additionally, our ConvNeXt-S–based PsychoNet required only a minimal modification to the ResNet-oriented Phasor Block (see below), yet still achieved performance close to ConvNeXt-S. This shows PsychoNet is compatible with modern CNN architectures, even under a conservative integration. More targeted adaptations of the Phasor Block could likely yield stronger performance in future work. We have revised Section 4 to clearly justify our choice of baseline networks.
>
> The ConvNeXt adaptation is only briefly mentioned in Section 3 (Phasor Blocks) because the modification is minimal: we just replace the real convolution in our Phasor Blocks with a $7\times7$ depthwise convolution block to match how depthwise convolution is the main convolution layer in ConvNeXt. Full architectural diagrams for both Phasor Block types are already provided in Appendix Figure A.2.
>
> > Emphasis is put on the number of layers in PsychoNet models being lower than their respective baselines, yet parameter counts remain similar.
>
> As described in the above response, the purpose of our quantitative evaluation was to show PsychoNet scaling is significantly less depth-dependent than ResNet. Parameter counts were matched for fair comparison.
>
> >Line 451's claim that "This pipeline mimics intermediate abstractions used by the brain to separate feature extraction from higher cognition" requires neuroscience citations to support the biological plausibility.
>
> In our introduction we cited four neuroscience works [1-4], which show evidence for different abstract representations observed in primate brains, as part of the motivation of our work. We do not repeat these citations in Line 451, since it is part of the conclusion, but did revise the sentence to:
>
> *“This pipeline provides strong evidence that it may mimic intermediate abstractions used by the brain to separate feature extraction from higher cognition as suggested in previous neuroscience studies.”*
>
> [1] Rodrigo Quian Quiroga. Concept cells: the building blocks of declarative memory functions. Nature Reviews Neuroscience, 2012. doi:10.1038/nrn3251.
>
> [2] Nikolaus Kriegeskorte, Marieke Mur, Douglas A. Ruff, Roozbeh Kiani, Jerzy Bodurka, Hossein Esteky, Keiji Tanaka, and Peter A. Bandettini. Matching categorical object representations in inferior temporal cortex of man and monkey. Neuron, 2008. doi: 10.1016/j.neuron.2008.10.043.
>
> [3] Rodrigo Quian Quiroga, Leila Reddy, Gabriel Kreiman, Christof Koch, and Itzhak Fried. Invariant visual representation by single neurons in the human brain. Nature, 2005. doi:10.1038/nature03687.
>
> [4] Lynn Le, Paolo Papale, K. Seeliger, Antonio Lozano, Thirza Dado, Feng Wang, Pieter R. Roelfsema, M. van Gerven, Ya˘gmur G¨uc¸l¨ut¨urk, and Umut G¨uc¸l¨u. Monkeysee: Space-time-resolved reconstructions of natural images from macaque multi-unit activity. Neural Information Processing Systems, 2024

---

> > ### Author Response · Authors · 2025-11-25
> > **Reponse to review oc5W (part 3)**
> >
> > >ResNet270 shows worse performance than ResNet152 on the same datasets, which is unexpected. Do you have a possible explanation?
> >
> > ResNet-270 outperforms ResNet-152 on ImageNet-1K and ImageNet-100, while it underperforms it on CIFAR-10 and CIFAR-100. CIFAR contains much smaller images ($32\times32$ opposed to $224\times224$ of ImageNet) and the extreme depth of ResNet-270 likely makes it poorly suited to such low-resolution inputs. In contrast, the much shallower PsychoNets are less affected which is an additional, albeit small, advantage of our approach. We have revised our discussion in Section 4 to include these details.
> >
> > >Figure 7 lacks interpretation of why imaginary features activate on whole objects rather than object parts like real features.
> >
> > This appears to be a unique emergent property of our Phasor Blocks. We recognise that further work is required to understand it, but it appears to be a helpful property, demonstrating our module captures both global context and highly localized object parts.

---

### Official Review · Reviewer_vJ9D · 2025-10-30

**Soundness:** 4
**Presentation:** 3
**Contribution:** 3
**Rating:** 8
**Confidence:** 2

**Summary:**

This paper introduces a new biologically-inspired algorithm for augmenting existing CNN-based vision models — PsychoNet. Briefly, this method enables models to translate low-level spatial information into high-level semantic information in the frequency domains using a suite of tools primarily based on 2-dimensional FFT. One key promise of this approach is the ability to represent semantic information in a more global, biologically-rooted way, in the frequency domain.
In general, I quite enjoyed this paper and felt like it was quite ‘dense’ in many respects. While the authors do a good job of laying out the core ideas, I think the ICLR audience would benefit from more scaffolding particularly on the human vision topics. Another consequence of the density of information is that several key ideas and details are bundled in the supplemental information section rather than the main text.
Overall, I think with some more scaffolded exploration of the background literature with some additional analyses I suggest below. This has the potential to be a strong contribution to ICLR.

**Strengths:**

* A clear novel architecture deeply grounded in studies on biological vision
* Clear technical explanations of the modeling choices, consequences of ablations, and also some cost-benefit tradeoffs w.r.t. compute in the supplementary materials.
* I appreciate the focus on building better models that aren't primarily based on scaling datasets and using standard transformer-based models.
* Potentially useful for downstream applications in the realm of human/primate vision.

**Weaknesses:**

* While I generally find the saliency map-based findings compelling in showing that PsychoNets acquire semantic information more efficiently and earlier relative to CNNs, I found the lack of non-accuracy based empirical comparisons lacking.  For example, if a key claim is that PsychoNet is more aligned with human vision, we should expect it to be more aligned with humans on key failure cases for CNNs including shape bias judgements, and the actual frequency code representations should also be predictive of human neural responses (say on open fMRI datasets like THINGS, etc.) I think some experiments clearly laying out the contributions of this modeling approach beyond visualizations and accuracy is needed for this to be a valuable contribution for the field.
One could imagine reporting the effects of the ablations currently presented (Fig 6) on these related benchmarks.
* There needs to be more background on psychovisual codes, not just on metrics used to capture these codes, especially given the audience. I think unpacking some of the ideas from Saadane et al., 1998 might be sufficient.
* Figure 5 a should be presented in a larger resolution with clearer font
* While from visual inspection it does appear that models do learn to recognize semantic parts, again having grounded metrics, with respect to annotations might be valuable.

**Questions:**

N/A. Refer to earlier sections!

---

> ### Author Response · Authors · 2025-11-25
> **Response to review vJ9D**
>
> We sincerely thank the reviewer for their encouraging feedback and careful assessment of our work. We also greatly appreciate their constructive comments, which were very helpful in guiding improvements to the paper.
>
> > the ICLR audience would benefit from more scaffolding particularly on the human vision topics
>
> We agree that this would allow broader appeal of our work and have added expanded appendices on the psychovisual and Fourier background material, including a more thorough review of the foundation work of Barba and his group on psychovisual analysis from the 1990s and the decade after (Appendix A.1). We have also added a portion of this additional information into the main text in the Background section as requested by the reviewer.
>
> > I think some experiments clearly laying out the contributions of this modeling approach beyond visualizations and accuracy is needed
>
> We thank the reviewer for pointing out possible experiments on prediction of human neural responses, especially suggestions of the THINGS fMRI dataset. We believe that this could be a very promising future research avenue to validate the psychovisual responses of SVCs with real human data. The THINGS concepts and images subset also appears to be a good fit as well for this purpose. In the meantime, we are endeavouring to provide additional non-accuracy based empirical evidence where possible in the revision to provide additional key points of alignment with human vision of PsychoNet variants, including a new Appendix C.3 showing the clustering of the features within PhasorBlocks by looking at their most significant eigenmodes in the complex domain, which tends to provide more evidence of the semantic nature of PsychoNet and  ImageNet-S50 experiments on an additional metric, see response further below.
>
> > There needs to be more background on psychovisual codes
>
> We thank the reviewer for this suggestion. We have extended the overview of psychovisual coding in Section 2, as well as added an expanded psychovisual Appendix A.1 that covers 12 additional works and motivations of Barba’s group in this area from the 1990s. We hope that this improves motivation and creates a better foundation for our work.
>
> > Figure 5 a should be presented in a larger resolution with clearer font
>
> We have improved this figure as requested in the revision.
>
> > While from visual inspection it does appear that models do learn to recognize semantic parts, again having grounded metrics, with respect to annotations might be valuable
>
> We are proposing the use of ImageNet-S50, which is subset of ImageNet-1K that has full object segmentations of a limited subset of ImageNet-1K classes, by reporting an overlap metric (e.g. mIoU or sensitivity) of regions detected by Phasor Block salience maps with respect to these segmentation masks on a subset of these labelled images (~50 images). We hope to add these as an appendix section to provide additional quantitative evidence as requested by the reviewer before the end of the Author-Reviewer discussion period.

---

> > ### Comment · Reviewer_vJ9D · 2025-11-25
> > **Response to authors**
> >
> > Thank you for the helpful responses. I will wait for the final results in the appendix.
> > I found most of the responses quite helpful.

---

### Official Review · Reviewer_1AWe · 2025-10-30

**Soundness:** 2
**Presentation:** 2
**Contribution:** 2
**Rating:** 2
**Confidence:** 3

**Summary:**

This work introduces Semantic Visual Coding (SVC), a learn frequency domain representation that introduces explicit psychovisual abstraction into CNNs The introduction of SVC is motivated from the perspective of producing disentangled representations to provide a more natural foundation for structured reasoning. SVC works by learning band-limited filters that encode semantics as distinct regions of the Discrete Fourier Transform. SVC is incrporated into some well-known CNN architectures and compared to its standard counter part. Activation maps and filter are visualized.

**Strengths:**

1. Nice illustration that visualize the methodology and results
2. Interesting link to biologically inspired vision.

**Weaknesses:**

1. The contribution of the work is unclear. There seems to exist a great deal of works that focus on transforming latent representations into the frequency domain [1, 2, 3, 4]. Some of these works are mentioned, other are not. However, the difference between SVC and existing works is not properly explained, and it is unclear what the methodological contributions are.

2. The experiments are poorly motivated and not properly evaluated.

(a) It is unclear what the purpose of the image classification experiments in Table 1 is. I interpret this sentence "Since we hypothesise that SVC should handle high-level processing, we stop increasing Phasor Blocks depth after Psycho-B/ResNet-101 to see if it can replace the role of late spatial layers (the width of existing layers are increased to compensate for parameter size.)" as the motivation for the reduction in layers. But the the motivation for the chosen baseline networks is unclear. The ResNet152 and 270 are rarely used, the ResNet18 and and resNet50 are much more common. The poor performance of ResNet152 and 270 on CIFAR10 is most likely due to low amount of samples compared to parameters. For the ConvNext-S, the performance difference is unclear.

(b) The activation maps and filter visualization are nice, but the analysis is highly qualitative. Without any baselines to compare against or quantitative measures, it is unclear what these results are actually demonstrating.

3. Comparison to existing works is missing. Without any comparison to other works, it it is difficult to asses the usefulness of SVC. There seems to be many alternatives that could be used or adapted for comparison [1, 2, 3, 4].

- [1] Lin et al., Deep Frequency Filtering for Domain Generalization, CVPR 2023
- [2] Chi et al., Fast Fourier Convolution, NeurIPS 2020
- [3] Rao et al., Global Filter Networks for Image Classification, Neurips 2021
- [4] Huang et al., Adaptive Frequency Filters As Efficient Global Token Mixers, ICCV 2023

**Questions:**

1. Concretely, in what ways does SVC differ from existing works in the literature like [1, 2, 3, 4]?
2. What benefits does SVC bring compared to existing works?
3. What quantitative measure can illustrate these benefits?
4. Can these benefits be shown experimentally?

- [1] Lin et al., Deep Frequency Filtering for Domain Generalization, CVPR 2023
- [2] Chi et al., Fast Fourier Convolution, NeurIPS 2020
- [3] Rao et al., Global Filter Networks for Image Classification, Neurips 2021
- [4] Huang et al., Adaptive Frequency Filters As Efficient Global Token Mixers, ICCV 2023

---

> ### Author Response · Authors · 2025-11-25
> **Response to review 1AWe (part 1)**
>
> We sincerely thank the reviewer for their thoughtful feedback and apologise that our contribution was unclear. We'd like to stress that our work is the first exploration of integrating psychovisual-style abstraction into deep learning models. Our approach yields far more interpretable and structured reasoning than standard CNNs, and also introduces a novel form of frequency-domain representation learning comprising naturally emergent sparse encodings of semantic features.
>
> > The contribution of the work is unclear / the difference between SVC and existing works is not properly explained, and it is unclear what the methodological contributions are / Comparison to existing works is missing
>
> We agree that although works [1-3] were already reviewed, the current frequency domain background section (Lines 129-134) could provide additional detail about works [1-4]. While SVC shares some of the basic operation of frequency domain filtering with prior work, SVC is a contribution to representation learning in the frequency domain. To the best of our knowledge, it is first to explore using the space for high-level semantic abstraction in the later stages of deep learning architectures. [1] uses frequency filtering to modulate features to improve domain generalization performance, rather than as a representational module or enabling interpretability. [2-4] are all performance-focused works which use frequency filtering to efficiently compute global spatial convolution via the Convolution Theorem.  [2] does so to incorporate global context within a multibranch/scale CNN network, while [3,4] uses it to implement token mixers for transformer-style models. Future work will see SVCs extended to transformers as well, while this current work is the first introduction of such data-driven visual codes and evidence it enables interpretable semantic reasoning in image classifiers. We thank the reviewer for pointing out [4], and have revised Section 2. Frequency Domain Learning accordingly.
>
> > The experiments are poorly motivated and not properly evaluated
>
> We designed our experiments to demonstrate that psychovisual processing can be meaningfully realised in deep learning and to assess whether it offers advantages for transparent reasoning. Image classification was chosen because it strongly relies on semantic feature extraction, which CNNs typically obtain through deep hierarchical layer stacks. However, these stacks often produce diffuse, blob-like activations with weak semantic specificity and limited interpretability. As expanded upon below, our results show that SVC add meaningful high-level abstractions and PsychoNet interpretable, psychovisual-like processing.
>
>
> > It is unclear what the purpose of the image classification experiments in Table 1 is / The motivation for the chosen baseline networks is unclear.
>
> We note that although only ResNet152 and 270 results are reported in Table 1, we had evaluated ResNet sizes 50, 101, which were included in Figure 5(a) and Table A.8/9. We have updated Table 1 to now include all four ResNet results.
>
> A common characteristic of newer and more advanced CNN architectures like ConvNeXt is their reduced dependence on increasing layer depth for scaling compared to older ResNets. Our results show that SVC similarly reduces the depth-dependence of ResNet, supporting that its frequency domain abstractions replace additional high-level processing from adding further spatial layers, it still achieved close performance with ConvNeXt-S with only a minimal modification to the ResNet-oriented Phasor Block using depthwise convolutions. This shows PsychoNet is compatible with modern CNN architectures and more targeted adaptations of Phasor Block may yield stronger performance in future work. We have revised Section 4 to clearly justify our choice of baseline networks.
>
>
> > The poor performance of ResNet152 and 270 on CIFAR10 is most likely due to low amount of samples compared to parameters
>
> While it is true that ResNet-152/270 have large parameter count compared to the number of samples in the CIFAR dataset, their counterpart PsychoNet models (Psycho-L/H) also have matching parameter count but perform better. This is likely as the extreme depth of the large ResNet models is excessive for the small $32\times32$ resolution of CIFAR images while the much shallower PsychoNets are less effected. This is an additional, albeit small, advantage of our approach. Section 4 has been revised to include this detail.

---

> > ### Author Response · Authors · 2025-11-25
> > **Response to review 1AWe (part 2)**
> >
> > > The activation maps and filter visualization are nice, but the analysis is highly qualitative. Without any baselines to compare against or quantitative measures, it is unclear what these results are actually demonstrating.
> >
> > Our visualisations and activations maps provide strong evidence that PsychoNet achieves psychovisual-like processing for the first time in deep learning. In Figure 5(b) PsychoNet’s spatial layers consistently focus on meaningful object parts, producing a structured separation between low and high-level processing. In contrast, the ResNet baseline showed diffuse, semantically vague activations. This comparison suggests that PsychoNet yields clearer and more semantically grounded reasoning signals than a standard CNN. Figures 6 and 8 further show that SVC learns sparse frequency domain abstractions which (1) organise object parts by scale between the different sub-bands and (2) encodes specific selections of parts per filter. These abstractions inform the model’s classification decisions, suggesting that these frequency-domain representations provide an interpretable basis for model reasoning.
> >
> > While these semantic reasoning findings are mostly qualitative, we believe that our initial study provides the first work to demonstrate a psychovisual approach in later stages of deep learning models. Developing and assessing alignment between learned parts and semantic concepts will be an important focus of future studies.

---

### Official Review · Reviewer_XNzN · 2025-11-01

**Soundness:** 3
**Presentation:** 3
**Contribution:** 2
**Rating:** 4
**Confidence:** 3

**Summary:**

This paper presents a two-stage hybrid architecture for computer vision that separates low-level feature extraction from high-level abstract reasoning. The first stage employs spatial layers to identify semantically meaningful object parts, augmenting real-valued features with complex-valued counterparts. The second stage transforms these features into the frequency domain using a Fast Fourier Transform, where a dedicated module performs the final classification using learned, sparse, band-limited filters. This architectural approach differs from some prior works that use the frequency domain to improve the efficiency of spatial convolutions or as an integrated global mixer. Here, the frequency domain is used as a distinct final stage to replace the deep spatial layers typically responsible for high-level reasoning. The entire framework is end-to-end differentiable, as all its components, including the Fourier transform and complex-valued convolutions, have well-defined gradients that allow for standard backpropagation-based training. According to the results, the model demonstrates improved interpretability and a reduced dependency on layer depth, though these benefits are accompanied by marginal performance gains over established baselines and a significantly higher computational cost. The work's contribution is therefore positioned as an exploration of an alternative, more transparent model design that bridges spatial feature extraction with frequency-domain abstraction.

**Strengths:**

The strengths of this work are centered on its architectural design, which aims to improve model interpretability, and the systematic experiments conducted to support its claims. The paper provides qualitative evidence through activation map visualizations suggesting that the framework successfully separates processing stages: early spatial layers learn to identify distinct, semantically meaningful object parts, while the subsequent frequency-domain module performs abstraction and reasoning on these parts. This separation is presented as a more transparent alternative to the entangled computations in deep, homogeneous CNNs.

I think:
1. The framework is explicitly designed to create a more interpretable processing pipeline by separating low-level feature extraction in the spatial domain from high-level reasoning in the frequency domain.

2. The results indicate that the proposed models can achieve comparable or slightly improved performance with significantly fewer layers than their deep ResNet baselines, suggesting that the frequency-domain module effectively handles the high-level processing that would otherwise require additional spatial layers.

3. The paper introduces the Phasor Block, a component whose design can be considered a notable contribution. Instead of adopting a computationally expensive, fully complex-valued network, the Phasor Block serves as a lightweight module that uses standard real-valued operations to generate complementary imaginary features just before they are needed for the Fourier transform. This represents a practical engineering solution to enable more expressive frequency-domain filtering without the full overhead of a complex-valued architecture.
4. The architecture is somewhat grounded in principles of psychovisual processing, providing a clear theoretical motivation for its design choices, particularly the use of coronal frequency bands for semantic abstraction.
5. The authors conduct extensive ablation studies to isolate and validate the contributions of their proposed components, such as the Phasor Blocks and Spectral Branches, adding rigor to their architectural claims.

I think the interpretability is pretty cool, albeit only justified qualitatively.

**Weaknesses:**

Despite its strengths in interpretability and design, the work has several weaknesses, primarily related to practical applicability and architectural complexity. The most significant drawback is the trade-off between computational cost and performance. The proposed models incur a substantial increase in computational overhead (FLOPs) compared to their baseline counterparts, yet the resulting improvements in classification accuracy are marginal at best, and in some cases, performance slightly degrades. This unfavorable trade-off makes the framework less compelling for applications where efficiency and predictive power are the primary concerns.

1. The models require significantly more FLOPs than the baselines they are compared against. The authors attribute this to the need for higher-resolution feature maps to support the frequency analysis and the use of complex-valued operations that are not highly optimized in current deep learning libraries, which poses a serious barrier to practical deployment.
2. The reported improvements in top-1 accuracy on benchmark datasets like ImageNet are minimal, often less than half a percentage point. Given the large increase in computational requirements, these small gains do not present a strong case for adopting the architecture based on performance alone.
3. Unlike modern architectures such as ResNet and Vision Transformers, which benefit from the simplicity and scalability of stacking homogenous blocks, the proposed framework is a heterogeneous, multi-stage pipeline. This design introduces significant architectural complexity by combining standard convolutional layers, specialized Phasor Blocks, a non-parametric FFT step, and frequency-domain filtering modules. This complexity makes the model less straightforward to scale and modify compared to simply adding more identical blocks.
4. The core concepts leveraged in the paper, such as frequency-domain analysis, complex-valued networks, and biologically-inspired architectures, are all pre-existing areas of research. The contribution can therefore be viewed as a specific and thoughtful synthesis of these ideas rather than the introduction of a fundamentally new technique, which may limit its perceived impact in a crowded field.
5. The framework is exclusively evaluated on image classification tasks. Its effectiveness on other critical computer vision tasks that require dense spatial predictions, such as object detection or semantic segmentation, remains unevaluated. It is unclear how the proposed abstraction in the frequency domain would perform on tasks where preserving precise spatial information is paramount.

Overall, my concerns can be broadly divided into two classes -- first being the improvement relative to FLOP, the second being novelty. Hopefully the authors can provide a more extensive justification.

**Questions:**

1. You propose the Phasor Blocks for introducing complex-valued features from real-valued inputs. However, their specific internal architecture is not fully justified.

-- What was the design process for the Phasor Blocks? Did you experiment with alternative, perhaps simpler, methods for generating imaginary components, such as a basic 1x1 convolution to project features into a complex space?

-- Why was the specific combination of depthwise and pointwise convolutions chosen? The paper states it's to encourage cross-channel interactions without interfering with spatial relationships, but it would be helpful to see an ablation study comparing this design to other methods.

2. The framework replaces the later stages of a CNN with the frequency-domain pipeline. This choice seems critical to the entire hypothesis but is based on empirical results. Is there a more principled way to determine the optimal depth for this transition, or is it purely a hyperparameter to be tuned for each base architecture?

3. The SVC module partitions the frequency spectrum into three fixed, disjoint radial bands. This seems to contradict the goal of a fully data-driven representation. Would a learnable partitioning scheme, where the model could adapt the frequency band boundaries, lead to better performance or even more specialized filters?

4. The paper repeatedly claims that the SVC module performs "high-level processing and reasoning." However, the evidence shows that it learns to encode selections of object parts. What reasoning is there?

5. The evaluation is confined to image classification on ResNet and ConvNeXt backbones. How do you expect this architecture to perform on dense prediction tasks like semantic segmentation or object detection, where precise, high-resolution spatial information is crucial for the final output? The frequency-domain abstraction inherently discards some spatial localization.

---

> ### Author Response · Authors · 2025-11-24
> **Response to review XNzN (part 1)**
>
> We sincerely thank the reviewer for their thoughtful and constructive feedback. We would like to first emphasise that the primary goal of this work is not to improve accuracy or computational efficiency, but to introduce a psychovisual-inspired framework that enables semantic abstraction in the frequency domain—a capability that is absent in existing CNNs.
>
> > The proposed models incur a substantial increase in computational overhead (FLOPs) compared to their baseline counterparts, yet the resulting improvements in classification accuracy are marginal at best … This unfavorable trade-off makes the framework less compelling for applications where efficiency and predictive power are the primary concerns
>
> Most of the current computational overhead arises from the proposed Phasor Block, whose complex-valued operations are designed to perform spatial-to-frequency conversion before SVC. Although improving the efficiency of complex-valued operations would certainly benefit our framework, this is a broad and active research topic that lies outside the scope of our core contribution. The higher-resolution feature maps used to support $14\times14$ SVC filters also increase FLOPs, but we chose this configuration to preserve architectural simplicity since our focus was on evaluating and analysing SVC behaviour. For the camera-ready version, we are exploring real-to-complex variants that replace the Phasor Block with simpler real-valued layers to reduce computational cost, although it may reduce the interpretability of spatial activations. We are also testing reducing internal feature map-resolution by instead extracting high-frequency sub-bands for SVC from earlier network stages. Furthermore, in domains where data is natively represented in the frequency domain—such as medical imaging modalities like MRI, overhead from complex values would largely disappear given the costly spatial-to-frequency conversion is no longer required, making PsychoNet even more computationally attractive. Overall, PsychoNet demonstrates strong compatibility with modern architectures, achieving accuracy comparable to state-of-the-art models such as ConvNeXt while providing interpretable psychovisual codes. This interpretability is particularly valuable for high-stakes medical imaging applications, where transparent model behavior is increasingly required by regulatory bodies such as the FDA. Its potential efficiency when operating directly on frequency-domain inputs further underscores its promise for MRI-relate application.
>
> > This design introduces significant architectural complexity by combining standard convolutional layers, specialized Phasor Blocks, a non-parametric FFT step, and frequency-domain filtering modules. This complexity makes the model less straightforward to scale and modify compared to simply adding more identical blocks.
>
> We clarify that standard convolutional layers are not part of the architectural design we introduce; they belong only to the ResNet backbone, which can be replaced by any CNN-based backbone. PsychoNet extends a backbone such as ResNet by replacing its deeper feature-extraction blocks with our proposed module (Phasor Block + SVC). Thus, unlike ResNet blocks that are simply repeatedly added for depth-based scaling, our block serves as a high-level processing unit that can be inserted at end of any CNN backbone to perform final semantic reasoning in the frequency domain. Regarding scalability, our design does not require stacking additional complex blocks. Scaling is achieved simply by increasing the number of filters within the Phasor Block and SVC, as demonstrated in our experiments. This avoids the problems seen when scaling ResNets by adding more repeated blocks—larger ResNets tend to overfit under small sample sizes and oversample at low input resolutions.

---

> > ### Author Response · Authors · 2025-11-24
> > **Response to review XNzN (part 2)**
> >
> > >The core concepts  leveraged in the paper, …, are all pre-existing areas of research. The contribution can therefore be viewed as a specific and thoughtful synthesis of these ideas rather than the introduction of a fundamentally new technique
> >
> > We respectfully disagree and believe this is the first work to investigate high-level semantic abstraction directly in the frequency domain within a psychovisual-inspired deep learning framework. While neuroscience studies have indicated that human visual decisions rely on structured, abstraction-driven processing, existing biologically motivated deep learning models primarily adapt low-level mechanisms like receptive fields. Existing frequency-based methods likewise focus on orthogonal goals, mainly reparameterizing spatial convolutions for efficiency, incorporating global context, or improving domain generalization; Section 2 has been revised to better clarify how these works differ from ours.  In contrast, our SVC mechanism, currently applied to CNNs, learns high-level representations in the frequency domain that enable structured separation between low and high level processing. These abstractions can additionally be directly visualised and linked to meaningful object parts that drive classification decisions, providing a more interpretable view of model reasoning than conventional CNNs. Our Phasor Block also introduces a novel mechanism for generating complementary complex-valued features (Appendix A.3) to process spatial signals in the frequency domain—a capability absent from prior work. Together, we believe these components represent a fundamentally new direction for integrating psychovisual reasoning into deep learning.
> >
> > > Its effectiveness on other critical computer vision tasks that require dense spatial predictions, such as object detection or semantic segmentation, remains unevaluated. It is unclear how the proposed abstraction in the frequency domain would perform on tasks where preserving precise spatial information is paramount.
> >
> > We thank the reviewer for their insight. Indeed, an important aspect is how the SVCs relate to semantic segmentation and how the abstraction in the frequency domain on more fine grain spatial tasks would be beneficial and is a current priority within our group, particularly in MRI applications, but requires significant time and resources to complete. However, we believe that the discovery of SVCs and particularly the psychovisual significance of our findings are important to pass onto community while this work is completed in the coming several months.

---

> > > ### Author Response · Authors · 2025-11-24
> > > **Response to review XNzN (part 3)**
> > >
> > > > Questions.
> > >
> > > **1.** The DW layers of previous models promoted the mixing of feature maps originating originally from colour channels and subsequent features. We hope to achieve the same in the complex domain to repeatedly develop a phasor representation of the features including colour and leverage the 2D nature (the Argand plane) in clustering semantic imformation. We have added initial clustering of Phasor blocks in Appendix C.3 to better illustrate the clustering we found.
> > >
> > > **2.** The initial layers of CNNs model the visual cortex with its use of receptive fields. Neuroscience has shown that decision making is more psychovisual as was targeted by the works of Barba and his group (see Appendix A.1). The reviewer correctly points out that our hypothesis is that the later layers of the CNN perform this psychovisual analysis as indicated by the main result of our work - SVCs. Currently, the optimal depth of our models have been found empirically, but we believe it should be possible to determine this with further work in the detailed analysis of optimal SVCs for image tasks, noting that our network finds them as a data-driven exercise, more optimal SVCs should be possible by inducing a bias for SVCs and modelling them mathematically that should allow for predicting depth more concretely.
> > >
> > > **3.**
> > > We thank the review for the suggestion, this would indeed be a more optimal way of sub-dividing bands. Currently, we use the dyadic approach that is predicted by Barba’s work on psychovisual coding, but also mimics how it is done in the spatial domain as well.
> > >
> > > **4.** We clarify that our use of “high-level processing and reasoning” is not intended to refer to formal notions of reasoning, such as symbolic or logical reasoning. In our context, it denotes the semantic feature aggregation stage immediately prior to classification, which standard CNNs implement through repeated deep spatial layers. Our results indicate that SVC and our frequency domain subsume this stage of processing since PsychoNet was significantly less depth-dependent compared to ResNet for model scaling.
> > >
> > > Our SVC also provides a novel, interpretable view into this stage: as shown in Figure 8, it first organises meaningful object parts by scale and then encodes sparse, part-specific selections in each filter, making explicit the semantic information used for the final decision. While connecting these abstractions to formal reasoning frameworks is future work, we believe this structured, object part-based processing already represents a promising direction towards more transparent and human-aligned vision models. We will revise our paper to clearly clarify our notion of "reasoning" by the end of the review period.
> > >
> > > **5.** In addition to our response above on semantic segmentation, we would like to add that segmentation using Fourier domain and SVCs are more likely to preserve spatial features, as evidenced by spectral pooling [1], while providing global and local support within models through low and high frequency bands respectively, the latter of which is still done separately as different stages of state-of-the-art visions models, such as GFNet.
> > >
> > > [1] Oren Rippel, Jasper Snoek, and Ryan P. Adams. Spectral Representations for Convolutional Neural Networks. In Proceedings of the 28th International Conference on Neural Information Processing Systems - Volume 2, NIPS’15, pp. 2449–2457, Cambridge, MA, USA, 2015. MIT Press.

---

### Author Response · Authors · 2025-11-24
**Global Rebuttal (Part 1)**

We thank the reviewers for their time and constructive feedback. We have addressed all concerns through a revised manuscript now available on OpenReview and detailed responses to each reviewer. Below is a summary of our clarifications.

**Foundation Work Clarification:**
Our work is inspired by groundbreaking research conducted over the course of a decade from 1991 by French researchers led by Dominique Barba in understanding the human aspect of mammalian vision, i.e. the psychovisual capability of the human brain. They proposed psychovisual coding as an image quantizer that would retain important image information pertaining to its interpretation by the human vision system. This used psychophysical experiments to select optimal frequency sub-bands that preserved perceptually salient features, enabling image compression that was difficult for humans to distinguish. We have extended Appendix A.1 with additional background and several additional citations to better highlight their psychovisual contributions and this theory is important to understand our main contributions.

**Main Contributions Clarification:** We provide the first data-driven exploration of psychovisual code theory within a deep learning framework, yielding novel learned Fourier representations that offer evidence for an interpretable semantic reasoning stage in deep layers of CNNs where none have been available before. Our findings support this view through: (1) \textbf{Frequency domain filter visualizations} showing learned filters that resemble psychovisual codes predicted by prior theory. (2) \textbf{Activation map analyses} demonstrating semantically meaningful, part-focused activation responses aligned with psychovisual principles. (3) \textbf{Comparable classification performance}, indicating that data-driven visual codes can serve as a viable and interpretable representational basis for modern vision tasks. We believe that our psychovisual approach in creating vision models that simultaneously provide interpretable data-driven functions as output and solid performance overall will be important in majority of practical applications involving human vision/interaction (such as medical imaging) and could serve as a basis for better understanding such models in the future.

To better clarify the focus of our work, **we propose changing the title of our work** to *"Semantic frequency domain codes for vision representation learning"* if permitted.

**Semantic Reasoning Clarification:** SVC enables models to form semantically grounded internal representations that clarify how visual information guides predictions in frequence domain. Unlike standard CNNs, which exhibit blob-like activations, SVC promotes the emergence of meaningful visual-code filters that highlight human-interpretable object parts and form structured abstractions. PsychoNet consistently attends to semantically relevant features, offering clearer reasoning paths and closer alignment with visual neuroscience perspectives. Its value lies in providing a foundation for semantically interpretable, general-purpose visual reasoning beyond accuracy or efficiency alone, with future work aimed at expanding these learned codes to broader visual tasks.

**Experiment Design Justification:** Our work is, to our knowledge, the first to study high-level semantic abstraction directly in the frequency domain using a psychovisual-inspired deep learning framework. Natural image classification was chosen to provide a practical testbed for evaluating whether psychovisual-like filters with semantic reasoning can emerge. ResNet serves as a meaningful baseline due to its tendency to produce blob-like activations, allowing us to assess SVC’s advantages in efficient high-level feature processing with reduced depth dependence, and semantically structured activation patterns for semantic reasoning. Prior frequency-domain works mainly target parameterising spatial convolutions for efficiency, introducing global context mechanisms, or focus on domain generalisation; in contrast, SVC learns sparse, interpretable filters corresponding to object parts (Figure 8), enabling semantic reasoning rather than merely performance-oriented filtering. Because these works differ fundamentally in purpose, direct experimental comparisons would offer limited conceptual insight. Section 2 has been revised with more thorough summaries of prior frequency domain work and clearer explanationd of how they differ to SVC. Our current classification-only evaluation focuses on establishing the feasibility and benefits of psychovisual processing within deep learning, with extensions to broader tasks planned for future work.

---

> ### Author Response · Authors · 2025-11-24
> **Global Rebuttal (Part 2)**
>
> **Quantitative Results Justification:** While the quantitative performance differences to ResNet and ConvNeXt are modest, raw performance is not the primary objective of this work. Our goal is to demonstrate that SVCs can be learned and integrated into frequency-based deep networks to yield meaningful semantic abstractions. Compared to ResNet, PsychoNet reduces reliance on depth for extracting high-level features by replacing deeper spatial layers with frequency-domain representations, resulting in semantically meaningful activations. Compared to ConvNeXt-S—a modern, high-performing baseline—our PsychoNet variant achieves comparable accuracy with minimal architectural modifications. This shows that SVCs are compatible with contemporary network designs while offering enhanced interpretability and improved stability across data scales.
>
> **FLOPs and Complexity Justification:** Most of the additional FLOPs in our method stem from the complex-valued operations in the Phasor Block during spatial-to-frequency conversion, rather than from the SVC mechanism itself. The higher-resolution feature maps used to support $14\times14$ SVC filters also add to the overhead, but we chose this setup to keep the architecture simple for the time being and focus on assessing SVC behaviour, and we are currently testing more resolution-efficient variants for the camera-ready version. Although improving the efficiency of complex-valued networks is an important direction for future work, it is a separate research challenge and does not detract from our core contribution: introducing a data-driven SVC deep framework. In domains where data is naturally represented in the frequency domain (e.g., MRI), the associated computational overhead largely vanishes. The interpretable visual codes produced by SVC are particularly valuable in such contexts, where regulatory bodies such as the FDA increasingly require transparent and trustworthy model behavior for clinical deployment. Mention of this motivation has been added to our Introduction and Future Work sections.
>
> Individual responses to each reviewer below attempt to address these and other concerns in a detailed manner.

---

### Author Response · Authors · 2025-12-03

We would like to thank the reviewers again for their constructive feedback, which greatly helped us improve our work. We have further revised the manuscript and added some additional results, summarised below.

- Further addressed Reviewer oc5W's feedback that our work overemphasises the ResNet-based PsychoNet compared to the ConvNeXt-based:
	- Added explicit description of key architectural differences between the two PsychoNet types in Section 3. PsychoNet.
	- Added full filter visualisation and salience map results for the ConvNeXt-based PsychoDW model (Appendix C.1.1) identical to those presented for the ResNet-based Psycho-B in Section 4. These demonstrate that the psychovisual-like interpretable semantic reasoning we find PsychoNet enables in ResNet also extends to ConvNeXt, and corresponding discussion has been added to Section 4 too.
	- Extended results discussion for ConvNeXt-based PsychoNet in Section 4.

- Further addressed Reviewer XNzN's feedback regarding the high FLOPs of PsychoNet compared to their baselines (Weakness 1), by adding an ablation study (Appendix D.1.1) demonstrating much more FLOP-efficient models are possible. These present FLOP-efficient variants of Psycho-S/B which maintain close performance on ImageNet-100 while greatly reducing FLOP usage - matching the original ResNet-50/101 baselines.

- Further addressed Reviewer XNzN's concerns regarding the complexity of scaling PsychoNet (Weakness 3) by adding Table A.1 to Appendix B.2 presenting a succinct comparison of Phasor Block and SVC layer configurations for each PsychoNet model. This demonstrates the simplicity of PsychoNet's scaling approach, where we just increase the number of Phasor Block and SVC filters without adding new layers. We also added discussion about it and comparisons to ResNet's depth-based scaling in Section 4.

- Revised discussions pertaining to reasoning in our text throughout to reflect the "Semantic Reasoning Clarification" in our Global Rebuttal.

- Further addressed Reviewer oc5W's feedback that Phasor Blocks receive disproportionate attention in the main text by moving the second paragraph in Section 3. Phasor Block, which presents in-depth architectural details about Phasor Blocks, to Appendix B.3 alongside the extended Phasor Block architecture figures.

Finally, we still aim to include ImageNet-S50 results for Reviewer vJ9D for a camera ready version if given the opportunity.

---

### Meta-Review · Area_Chair_Fzgb · 2025-12-21

**Summary:**

The paper proposes SVC/PsychoNet: a hybrid CNN where late layers are replaced by learned frequency-domain (Fourier) band-limited filters to yield more “psychovisual” and interpretable semantic abstractions; evidence is mainly qualitative with modest classification gains and high compute.

Strengths
* Novel-ish architectural separation of spatial feature extraction vs. frequency-domain abstraction.
* Nice filter/saliency visualizations suggesting more part-based, structured representations.
* Reasonable ablations; authors improved ConvNeXt discussion and some clarity in rebuttal.

Weaknesses
* Contribution/novelty unclear vs. existing frequency-domain filtering/mixer work; insufficient direct comparisons.
* Central “reasoning/abstraction” claim is mostly qualitative; lacks strong quantitative metrics (e.g., segmentation/part overlap, shape bias, robustness, neural predictivity).
* Bad FLOPs/accuracy trade-off and added architectural complexity reduce practical appeal.
* Scope limited to classification; unclear behavior on dense tasks.

Overall, interesting exploratory direction, but not yet convincingly differentiated or empirically grounded enough (especially given cost) for acceptance.

**Reviewer Scores:**

n/a

---

### Decision · Program_Chairs · 2026-01-26

Reject